# The future North Atlantic jet stream and storm track: relative contributions from sea ice and sea surface temperature changes

Daniel Köhler[1], Petri Räisänen[2], Tuomas Naakka[2, 3], Kalle Nordling[2], and Victoria A. Sinclair[1]

[1]Institute for Atmospheric and Earth System Research/Physics, Faculty of Science, P.O. Box 64, University of Helsinki, 00014 Helsinki, Finland
[2]Finnish Meteorological Institute, Helsinki, Finland
[3]Department of Meteorology and Bolin Centre for Climate Research, Stockholm University, Stockholm, Sweden

**Correspondence:** Daniel Köhler (daniel.kohler@helsinki.fi)

**Abstract.** Using a novel set of coordinated simulations from four different models, the response of the wintertime (December – February) North Atlantic jet stream and storm track to prescribed sea surface temperature increases and sea-ice loss is analysed and the underlying physical mechanisms investigated. Three out of the four models show a southward shift of the upper-level jet stream with an increase in jet speed over Europe, where the contribution of sea surface temperatures dominates over the effects of sea-ice loss. However, the remaining model lacks the increase in jet speed over Europe, which originates from opposite responses of similar magnitude due to the future sea surface temperatures and sea-ice cover. The jet stream responses are primarily driven by the change in the meridional temperature gradient and, as a consequence, baroclinicity. At the same time, momentum flux convergence acts as a secondary amplifying and dampening factor. The same three models see a significant eastward shift of the extratropical cyclone track density, which is equally driven by changes to sea surface temperatures and sea ice cover. A consistent feature across all models is a decrease in the frequency of extratropical cyclones in the Mediterranean. The responses of extratropical cyclones to future sea-ice cover and sea surface temperatures do not exceed the inter-model climatological differences. Notable differences in the future response of the jet stream and storm track occur, and thus considerable uncertainty remains in how the European climate will respond to a warmer climate.

## 1 Introduction

Global warming due to rising greenhouse gas concentrations is accompanied by a warming of the average sea surface temperatures (SSTs) and loss of sea ice cover (SIC). In particular, the Arctic is severely impacted as it is currently warming up to 4 times faster than the global average, a phenomenon known as Arctic Amplification (Rantanen et al., 2022), and is experiencing dramatic sea-ice loss (Simmonds and Li, 2021). The sea-ice loss strongly contributes to a locally enhanced warming and moistening of the lower troposphere (Screen and Simmonds, 2010). However, the effects of sea-ice loss on the mid-latitude circulation is a topic of intense debate (Smith et al., 2022; Screen et al., 2022; Ye et al., 2023).

Mid-latitude weather is characterised by extratropical cyclones (ETCs), which are organised into storm tracks on climatological timescales. Another prominent atmospheric feature in the mid-latitudes is the eddy-driven jet stream, which acts as a guide for ETCs. It is critical to study the storm track and the jet stream together, as they are closely connected (Athanasiadis

et al., 2010; Ronalds and Barnes, 2019; Ye et al., 2023). The response of the North Atlantic jet stream to a warmer climate
shows considerable inter-model variability over Europe (Zappa et al., 2018; Oudar et al., 2020). Moreover, future changes to
SSTs and SIC have been shown to have opposing influences on the jet stream. A poleward shift of the jet stream is associated
with rising SSTs, meanwhile, SIC loss leads to an equatorward shift (Barnes and Screen, 2015; Screen et al., 2018; Yu et al.,
2024). A similar pattern has been reported for the storm tracks (Yu et al., 2023). In particular, the North Atlantic jet response to
sea ice is highly uncertain (Screen et al., 2018), especially in the Northern Hemisphere winter (Simpson et al., 2014; Hay et al.,
2022). Furthermore, the sea-ice loss-related changes to the storm tracks are tightly linked to the jet stream changes (Ye et al.,
2023; Ronalds and Barnes, 2019), necessitating more studies simultaneously investigating jet stream and storm track changes.

A major source of uncertainty in future projections from fully coupled climate models is the differing amount of sea-ice loss
that different models predict (Notz and Community, 2020). Moreover, the region and magnitude of sea-ice loss also has been
shown to have a substantial impact on the atmospheric response. For example, links between sea-ice loss and an intensification
of the negative phase of the North Atlantic Oscillation on a climatological time scales have been identified by Screen (2017).
Using an intermediate complexity model, McKenna et al. (2018) showed moderate sea-ice loss in the Atlantic (Pacific) sector
leads to a negative (positive) Arctic Oscillation response, meanwhile extensive sea-ice loss in either sector leads to a negative
Arctic Oscillation response.

The response of the mid-latitude tropospheric zonal wind, which is closely related to the jet stream, is proportional to the
eddy momentum feedback (Smith et al., 2022; Screen et al., 2022). Smith et al. (2022) showed that models tend to underesti-
mate responses of mid-latitude tropospheric zonal wind due to changes in SIC when constrained by observations of the eddy
momentum feedback. It is important to note that the magnitude of the underestimation is strongly model-dependent. This limits
the attribution of mid-latitude changes to differences in sea-ice loss or model representation of the atmospheric interactions
between high-latitudes and mid-latitudes. To reduce the impact of different models' biases on the future response from future
projections, efforts are made to simultaneously analyse output from numerous models (Eyring et al., 2016; O'Neill et al., 2016;
Smith et al., 2019). However, this often limits studies to the use of multimodel means, which reduces the physical interpretabil-
ity of future climate responses. On the other hand, studies focusing on the physical mechanism tend to employ one model,
which have reduced accountability for modelling-related uncertainties (Levine et al., 2021; Dai and Song, 2020; Chemke et al.,
2019).

This study is part of the "Climate Relevant interactions and feedbacks: the key role of sea ice and Snow in the polar and
global climate system" (CRiceS) project. CRiceS aims to understand the role of the polar processes, such as feedback loops,
in polar and global climate. This includes quantifying processes that drive interactions and teleconnections between the higher
and lower latitudes. For this purpose, coordinated model experiments investigating the contributions of SST and SIC changes to
the future climate response are performed. The CRiceS simulations are following previous studies on the effect of sea ice loss,
which have performed model simulations with prescribed sea ice cover using coupled models (McCusker et al., 2017; Oudar
et al., 2017) and atmospheric general circulation models (AGCMs) (Deser et al., 2010; Smith et al., 2019). These simulations,
each encompassing a continuous 40-year simulation, are performed using four AGCMs (OpenIFS-43r3, EC-Earth3, CESM2,
NorESM2) which are all forced with the same prescribed SST and SIC for historical and multiple future climate conditions

(Naakka et al., 2024). The full set of simulations includes a baseline simulation with historical SSTs and SIC, future simulations where both SSTs and SIC are changed simultaneously according to different emission scenarios, and simulations where the SSTs and SIC are changed independently. Thus, the individual contributions of SIC and SST are obtained by leveraging the power of the full set of experiments, — discussed further in section 2.1. Prescribing either future SIC or SST while keeping the other at historical levels allows us to study the contributions of changes in SIC and SST in isolation, which is a limitation of fully-coupled climate simulations like CMIP6 (Eyring et al., 2016). An additional benefit of prescribed SIC and SST is reduced internal variability, which improves the detection of future climate signals. To further increase the likelihood of detecting statistically significant responses, the experiments representing future climate conditions use stronger warming scenarios for SST and SIC, compared to previous studies (Smith et al., 2019; Peings and Magnusdottir, 2014; Screen et al., 2012; Deser et al., 2010). Moreover, by analysing carefully designed simulations from a relatively limited number of models, rather than all models from the CMIP6 archive, leads to a reduction in the current uncertainties of the responses due to differences in the projected SIC and SST. This is enabled by high output frequency of a wide selection of atmospheric variables, which permits to examine the underlying physical mechanisms and identify structural differences in physical mechanisms of the response to projected SIC and SST across the selected models,

This work aims to determine the wintertime response of the North Atlantic jet stream and storm track to changes in sea ice cover and sea surface temperatures and quantify their relative contributions to future climate conditions with simultaneous changes to SST and SIC. Specifically, the following questions are investigated:

- How well do the four models agree on the climatologies of the historical climate simulation when SSTs and sea ice cover are identical? The focus is on the winter North Atlantic jet stream, storm track and individual extratropical cyclones.

- Are there differences across models in the total future climate response when SSTs and sea ice cover are changed simultaneously?

- What are the relative contribution of SST and sea-ice cover changes to the total future climate response for each individual model? In particular, do the contributions oppose or amplify each other?

- What are the physical mechanisms leading to the responses due to changed SSTs and sea ice cover?

The study is structured as follows. Section 2 presents the conducted model simulations and introduces the metrics used to study the North Atlantic jet stream and storm track. Section 3 compares the North Atlantic jet stream and storm track across models in current climate conditions. Section 4 addresses the response of the North Atlantic jet stream in future climate and the contribution of SST and SIC. The physical mechanisms underlying the responses are explored in Section 5. Section 6 focusses on the responses of ETCs, including the contribution of SSTs and SIC. Specifically, it investigates the storm track density and multiple important ETC specific metrics like lifetime, maximum intensity and more. The study is concluded in Section 7.

## 2  Data and methods

### 2.1  Models and simulations

This study utilises a set of simulations by the CRiceS consortium, consisting of atmosphere-only simulations by three Earth system models (EC-Earth3, CESM2, and NorESM2) and one general circulation model (OpenIFS-43r3). The conducted simulations were designed with the aim of studying the impacts of changes in sea surface temperatures (SSTs) and sea ice cover (SIC). This is achieved by running experiments prescribing different combinations of historical and future SIC and SSTs. While previous studies used multimodel means of SST and SIC to force their simulations (Smith et al., 2019; Deser et al., 2010), in the CRiceS simulations the SST and SIC boundary conditions are obtained from the Australian Earth system model ACCESS-ESM1.5 from the CMIP6 archive (Eyring et al., 2016). ACCESS-ESM1.5 produces an Arctic sea ice cover evolution for the historical period that is in reasonable agreement with observations and provides the best guess estimate for future SIC (Notz and Community, 2020). Monthly climatological means from 1950 – 1969 are used as historical boundary conditions for the seasonal cycle, which is annually repeated. Meanwhile, the future boundary conditions use output from 2080 – 2099 under either the shared socioeconomic pathway SSP 1-2.6 or SSP 5-8.5 scenario. The full simulation set consists of a Baseline simulation $BL$ (historical SST & SIC), two Future simulations $FT_x$ (future SST & SIC), two SST simulations $ftSST_x$ (future SST & historical SIC), and two SIC simulations $ftSIC_x$ (historical SST & future SIC) where $x$ is either SSP126 or SSP585 and denotes which socioeconomic pathway the boundary conditions are taken from. The simulation set is summarised in Table 1. Each simulation was run for 40 years + 1 year of spin-up, starting on the 1st of January. The variables related to dynamics and thermodynamics are saved as 6-hourly output, enabling more diagnostics to be computed offline. A more detailed description of the models and simulations, including a basic meteorological analysis of temperature, mean sea level pressure, and precipitation, is found in Naakka et al. (2024).

|  | SIC (Historical) | SIC (SSP 1-2.6) | SIC (SSP 5-8.5) |
|---|---|---|---|
| SST (Historical) | $\boldsymbol{BL}$ | $ftSIC_{SSP126}$ | $\boldsymbol{ftSIC_{SSP585}}$ |
| SST (SSP 1-2.6) | $ftSST_{SSP126}$ | $FT_{SSP126}$ | – |
| SST (SSP 5-8.5) | $\boldsymbol{ftSST_{SSP585}}$ | – | $\boldsymbol{FT_{SSP585}}$ |

**Table 1.** The experiment names of the CRiceS coordinated simulation set performed by OpenIFS-43r3, EC-Earth3, CESM2, and NorESM2. The rows correspond to the prescribed sea surface temperature SST, while the column correspond to the prescribed sea ice cover SIC. SSP refers to the shared socioeconomic pathways from O'Neill et al. (2016). This study uses the $BL$ and SSP 5-8.5 simulations (shown in bold).

The CRiceS simulation set bears close resemblance to the set-up used by the Polar Amplification Model Intercomparison Project PAMIP (Smith et al., 2019), yet a few notable differences exist. While PAMIP uses a 100-member ensemble of 1-year simulations, the CRiceS simulations are comprised of 40-year continuous simulations to enable analysis using a continuous time series. However, using PAMIP simulations, Peings et al. (2021) have showed that while the local thermal response to Arctic sea ice loss is very consistent across the different 100-member ensembles, the mid-latitude circulation response differs

significantly. Therefore, results need to be carefully interpreted when isolating the response of mid-latitude circulation to sea
ice loss at +2 K global warming. In the CRiceS simulations, the SST and SIC boundary conditions for SSP 1-2.6 correspond to
+1.82 K global warming and for SSP 5-8.5 correspond +4.4 K warming compared to the Baseline simulation. The SSP 5-8.5
simulations provide a stronger SIC and SST forcing (Fig. A1) compared to PAMIP simulations, which improves the signal
strength compared to internal variability, therefore facilitating a more robust detection of responses due to SIC and/or SST
changes.

This work analyses the 39 available complete Northern Hemisphere (NH) winters (DJF) using the SSP 5-8.5 $FT_{SSP585}$,
$ftSST_{SSP585}$, $ftSIC_{SSP585}$, and Baseline $BL$ simulation. The analysis is limited to the North Atlantic (NA) sector defined
as 95° W – 45° E and 20° N – 80° N with a focus on Europe (15° W – 35° E and 30° N – 70° N) given the high population
density in this region and the potential for large societal impacts if the jet stream and ETC characteristics change in the
future. The model output is vertically interpolated from model levels to isobaric surfaces from 1000 hPa to 50 hPa in 50
hPa intervals. The horizontal resolution is kept at the native resolution of each model. OpenIFS-43r3 and EC-Earth3 are run
with TL255 horizontal resolution (0.7°x0.7° at the equator) and 91 vertical model levels. NorESM2 has a longitude-latitude
resolution of 2.5° x 1.9°, while for CESM2 the resolution is 1.25° x 0.9°. NorESM2 and CESM2 are run with 32 model levels.
The selected models can be grouped into two families based on their atmospheric component. The Integrated Forecast System
IFS, developed by the European Centre for Medium-Range Weather Forecasts, is the basis for EC-Earth3 (IFS Cycle 36r4)
and OpenIFS-43r3 (Cycle 43r3). Similarly, the atmospheric component of NorESM2 was developed from the atmospheric
component of CESM2, namely the Community Atmospheric Model CAM6.

The responses are calculated as the difference of the December to February climatological means between a perturbed
simulation and the Baseline simulation $BL$. The mathematical formulation is as follows:

$$\Delta FT = FT_{SSP585} - BL,$$
$$\Delta SST = ftSST_{SSP585} - BL,$$
$$\Delta SIC = ftSIC_{SSP585} - BL.$$

(1)

Note the scenario subscripts are omitted in the name of the response as this study only uses the SSP 5-8.5 scenario. The
difference between the $FT_{SSP585}$ and Baseline climatology is referred to as Future response, denoted by $\Delta FT$. The response
to changes in sea surface temperature, $\Delta SST$, is calculated from the difference between the SST simulation $ftSST_{SSP585}$ and
Baseline simulation $BL$, and the sea ice response $\Delta SIC$ is calculated likewise. It is important to highlight that the summation
of $\Delta SIC$ and $\Delta SST$ does not result in $\Delta FT$. The differences arise from the lack of the effect of changed SSTs where sea
ice is removed and resulting non-linear interactions. Additionally, the internal variability also contributes to the differences
between $\Delta FT$ and $\Delta SST + \Delta SIC$. For a more in-depth analysis of the non-linear interactions in the CRiceS simulation set,
consult Naakka et al. (2024).

To test if the climatological means of perturbed experiments are statistically different from the Baseline mean climatology
for gridded data, a two-tailed t-test with the 39 seasonal means for DJF as input is performed. Subsequently, the significance

was controlled using false discovery rate (FDR) according to Wilks (2016). Furthermore, the consistency discovery rate (CDR) proposed by Peings et al. (2021) was applied to test for a significant consistent sign in the response. CDR testing in this study takes the following steps. 39 responses ($\Delta FT$, $\Delta SST$, $\Delta SIC$) from the seasonal DJF means. A subsample of 20 from 39 responses are drawn without repetition. The climatological average is calculated and the sign is recorded. The subsampling is repeated 1000 times. If 900 iterations agree on the sign, the sign of the response is considered significantly consistent. In the

Figures, shading indicates the responses which show CDR consistent response and areas without stippling show statistically significant responses with the FDR corrected t-test. Significant differences in ETC quantities described in Sect. 2.3 are detected using a Mann-Whitney U test with threshold p-value < 0.05.

## 2.2 Baroclinicity and momentum flux convergence

The strength of the NA jet stream is predominantly governed by two physical mechanisms. One is the thermal wind law,
which describes vertical changes in the wind speed due to a horizontal temperature gradient. Relevant for the jet stream is the negative meridional temperature gradient (e.g. temperature decreases towards the poles) in the troposphere, which leads to an increasing zonal wind with height. The other mechanism originates from eddy-mean flow interaction, whereby the momentum of atmospheric waves (eddies) is fed back to the jet stream (Eliassen and Palm, 1961; Hoskins et al., 1983).

The necessary metrics to study the mechanisms driving the NA jet stream require eddy-mean flow separation, which is a
common technique to study synoptic-scale atmospheric dynamics. This is achieved by low-pass filtering using a 21-weight Lanczos filter with a 10-day cut-off period. The longer timescale of 10 days compared to the more common 6 days is motivated by including breaking synoptic waves as part of the eddy flow (Rivière et al., 2018). For the filtering, the shoulder months (November and March) are added to the data set and then subsequently discarded for analysis. The low-pass filter is applied to the potential temperature $\theta$, and wind components $u$ and $v$. The eddy part of the wind components $u'$ and $v'$ is determined by
subtracting the low-pass filter field from the full field.

In isobaric coordinates, the horizontal temperature gradient is proportional to the potential temperature gradient, which is quantified by the meridional component of the baroclinicity vector $B_y$. The baroclinicity vector, $\boldsymbol{B}_s$, is given by

$$\boldsymbol{B}_s = -\frac{\nabla \bar{\theta}}{\sqrt{S}}, \text{ where } S = \frac{1}{h}\frac{\partial \bar{\theta}}{\partial p} \text{ and } h = \frac{R}{p}\left(\frac{p}{p_0}\right)^{R/c_p}, \tag{2}$$

as used by Cai and Mak (1990) and Schemm and Rivière (2019). $\boldsymbol{B}_s$ is calculated from the low-pass filtered potential tem-
perature $\bar{\theta}$. Additionally, the temperature gradient $\nabla \bar{\theta}$ is normalised by the stability $S$, which uses the scale height $h$, which includes the gas constant for air $R$, specific heat capacity of air at constant pressure $c_p$, and reference pressure $p_0 = 1000$ hPa. The zonal wind speed and the Eady growth rate are proportional to the meridional component of the baroclinicity $B_y$.

Multiple approaches have been developed to characterise the eddy-mean flow interaction and its effects on the jet speed, most notable being the Eliassen-Palm flux (Eliassen and Palm, 1961) and E-vectors (Hoskins et al., 1983), and many more
exist (Trenberth, 1986; Plumb, 1985). The Eliassen-Palm flux uses the zonal averages and quantifies eddies as deviations from the zonal average. On the other hand, E-vectors utilise all 3 spatial dimensions, where the eddies and mean flow are obtained by using temporal filters. Both methods are physically motivated by the transfer of momentum between eddies and mean flow.

Additionally, the mathematical form is identical for zonal averages (Trenberth, 1986). The present article uses the meridional component $F_\phi$ of the E-vectors, defined as

$$F_\phi = a \, \cos\phi \, (-u'v'), \tag{3}$$

where $a$ is the radius of Earth and $\phi$ is the latitude. The variables $u'$ and $v'$ symbolise eddy in terms of the horizontal wind components. Specifically, the momentum flux convergence

$$MFC = \left\langle \frac{1}{a \, \cos\phi} \frac{\partial F_\phi \, \cos\phi}{\partial\phi} \right\rangle \tag{4}$$

is calculated and results in the zonal mean, denoted by $\langle \cdot \rangle$. The momentum flux convergence $MFC$ is proportional to the acceleration of the zonal wind speed $u$ for frictionless motion, and quantifies the feedback of eddies on the jet. A similar metric was used by Smith et al. (2022) to assess the zonal wind speed response to changes in Arctic sea ice.

While the temperature gradient and eddy-mean flow interaction mechanisms are not directly quantitively comparable, they are tightly connected in the jet stream area. By comparing the sign of changes in baroclinicity and momentum flux convergence, a qualitative interpretation of compounding or opposing mechanistic effects on the responses of the zonal wind due to sea ice and SST changes is possible.

## 2.3 Extratropical cyclone tracking

Extratropical cyclones (ETCs) are objectively identified by the TRACK algorithm (Hodges, 1994 & 1999). The tracking is performed using mean sea level pressure, as it is available as direct model output for all four models. The mean sea level pressure is truncated to T63 resolution with wave numbers 5 and lower removed, from which local minima are identified as ETCs. Like Priestley et al. (2023), the individual ETCs are filtered according to multiple criteria: they have to be (1) mobile (travel at least 1000 km), (2) long-lasting (have a lifetime of at least 48 hours), (3) affect Europe (at least 48 hours within the European box as defined in Sect. 2.1)and (4) and occur in the NH winter (genesis date in DJF).

Multiple quantities are derived from the TRACK output for each entire ETC track satisfying conditions (1), (2), (3) and (4). Track count originates from the number of tracks, track duration is obtained from the number of timesteps, genesis latitude corresponds to the latitude at the first timestep, and latitudinal displacement is the difference in latitude between the last and first timestep. The mean speed is obtained by calculating the speed at each timestep from the change in coordinates and subsequently averaged over the track. The maximum intensity is the maximum of the negative pressure anomaly from the T63 mean sea level pressure within a single ETC track. For ETCs satisfying conditions (1), (2), and (4), the track density is computed using spherical kernels (Hodges, 1996) and estimates the likelihood that a given point is affected by an ETC.

## 3 Intercomparison of the Baseline simulation

This section compares the Baseline simulation across models. For readability, a detailed description of the Baseline simulation of OpenIFS-43r3 is given in each subsection. Subsequently, the key differences between the three remaining models and OpenIFS-43r3 are presented.

## 3.1 The horizontal perspective

The NA jet stream, identified using the 250-hPa zonal wind speed, originates at the North American east coast and ends over Europe. In OpenIFS-43r3 (Fig. 1a, colours), the NA jet stream is strongest ($36 \ \mathrm{m \ s^{-1}}$ and $40 \ \mathrm{m \ s^{-1}}$) over the western NA between 30 °N – 40 °N. Towards the east, the jet is narrower, weaker, and the jet core moves northward, commonly referred to as the tilt of the NA jet. The eddy-driven jet, identified via the 850-hPa zonal wind as in Woollings et al. (2010), is strongest ($9 \ \mathrm{m \ s^{-1}}$ to $12 \ \mathrm{m \ s^{-1}}$) in the Central Atlantic (Fig. 2a, contours), which is east of the upper-level maximum. Similar to the upper-level jet, the low-level jet shows a northward tilt from west to east. The structure of both the upper-level and lower-level NA jet stream in the OpenIFS-43r3 Baseline simulation is in good agreement with what is found in reanalysis (Fig. A2) and historical simulations from coupled climate models (Harvey et al., 2020), despite the Baseline simulation not being directly comparable to these as there is no interannual variability of SST and SIC.

The baroclinicity $B_y$ is diagnosed at $500 \ \mathrm{hPa}$. The maxima and minima in $B_y$ closely coincide with the maxima and minima in the 250-hPa zonal wind. In the NA (Fig. 1a, contours), the maximum in $B_y$ is located slightly northward of the upper level jet maximum, yet their respective shapes match closely. As was the case with the jet stream, $B_y$ shows a northward tilt and an eastward decrease in magnitude.

The ETC track density is tightly linked to the presence of baroclinicity and the jet stream. Three strong maxima are identified (Fig. 2a, colours), one over the eastern USA, another at the North American East coast, and the last one south-east of Greenland. Generally, the ETC track density is highest in regions of high baroclinicity and on the northern flank of the lower-level and upper-level jet stream, thus following the northward tilt across the NA. Further prominent features are elevated ETC track density in the Mediterranean and Baltic Sea. The ETC densities are in good agreement (not shown) with reanalysis (Gramcianinov et al., 2020) and coupled climate models (Priestley et al., 2020).

Generally, the Baseline simulation from EC-Earth3 (Fig. 1b & Fig. 2b) is similar to that from OpenIFS-43r3 when the zonal wind speed at $250 \ \mathrm{hPa}$ and at $850 \ \mathrm{hPa}$, baroclinicity, and ETC track density are considered. Minor differences are found in the ETC densities, which are 40 % higher in the north-eastern USA and mainland Europe compared to OpenIFS-43r3. NorESM2 and CESM2 have comparable Baseline simulations to OpenIFS-43r3 and EC-Earth3. However, there are multiple distinguishing features. Most notably, the low-level jet stream is considerably faster, oriented more zonally, and the maximum shifted towards the east (Fig. 2c,d, contours) in both NorESM2 and CESM2 compared to OpenIFS-43r3 and EC-Earth3. Furthermore, the upper-level jet stream speed reduces more gradually towards the east, which results in higher jet speed over Europe (Fig. 1c,d, colours) in NorESM2 and CESM2. Moreover, the ETC track density is higher extending from the North American East coast over the south-east of Greenland to Northern Europe (Fig. 2c,d, colours). Lastly, in NorESM2 and CESM2 the baroclinicity is lower outside the jet stream regions.

While NorESM2 and CESM2 share a similar pattern, distinguishing them from OpenIFS-43r3 and EC-Earth3, differences also exist between NorESM2 and CESM2. NorESM2 shows a 10 % stronger upper-level jet over Europe (Fig. 1c, colours) compared to CESM2 (Fig. 1d, colours). Another feature is in the ETC densities over Northern Europe, where CESM2 (Fig.

2d, colours) has higher values than NorESM2 (Fig. 2c, colours). This is related to higher ETC counts in CESM2 compared to NorESM2, which is discussed in Section 3.3.

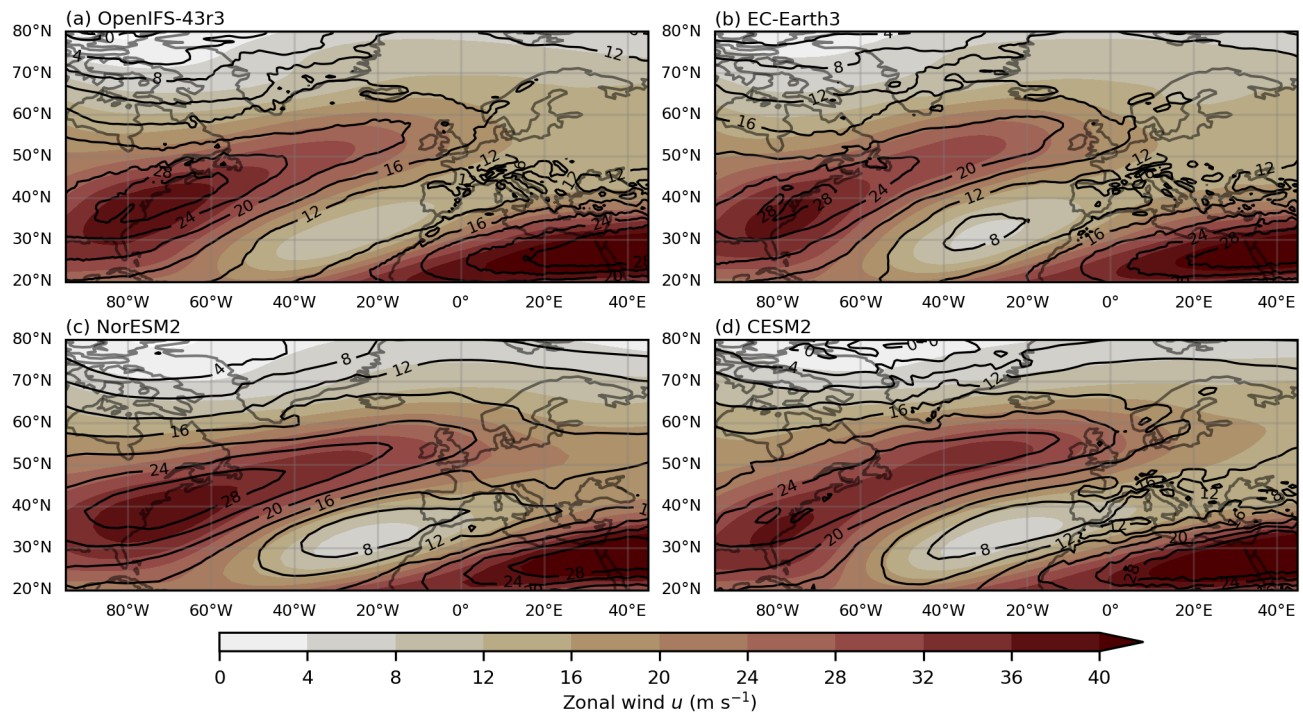

**Figure 1.** The 39-year DJF mean of zonal wind speed $u$ at 250 hPa (in colour shading, m s$^{-1}$) and baroclinicity $B_y$ at 500 hPa (in black contours, $10^{-4}$ s$^{-1}$) for (a) OpenIFS-43r3, (b) EC-Earth3, (c) NorEMS2, and (d) CESM2.

## 3.2 The zonal cross-section perspective

This study's primary focus is on Europe (30° N – 70° N, 15° W – 35° E) for multiple reasons. First, Europe is a densely population region where the jet stream and ETCs may lead to societal impact. Second, this is where the exit region of the NA jet is located and the left hand jet exit is an area of upper-level forcing that induces ascent, which is favourable for extratropical cyclone intensification. Third, the four models have shown differences in the zonal wind over Europe at 250 hPa (Fig. 1) and 850 hPa (Fig. 2). The cross-sections of the zonal mean are utilised in Figure 3, where the zonal wind speed $u$ (in colours), the

baroclinicity $B_y$ (in blue contours), and the eddy momentum flux convergence $MFC$ (in black contours) are displayed. This allows insight into the European jet properties across models.

The OpenIFS-43r3 Baseline cross-section is presented in Figure 3a. The edge of the subtropical jet stream located over Northern Africa (around 30° N and 200 hPa) is visible as high values of $u$. A second maximum (values between 16 m s$^{-1}$ and 18 m s$^{-1}$) in zonal wind speed at 250 hPa and between 50° N and 60° N is associated with the jet exit region of the NA

jet stream. An area of increased baroclinicity $B_y$ (600 hPa to 400 hPa) is found below and north of the NA jet maximum. In

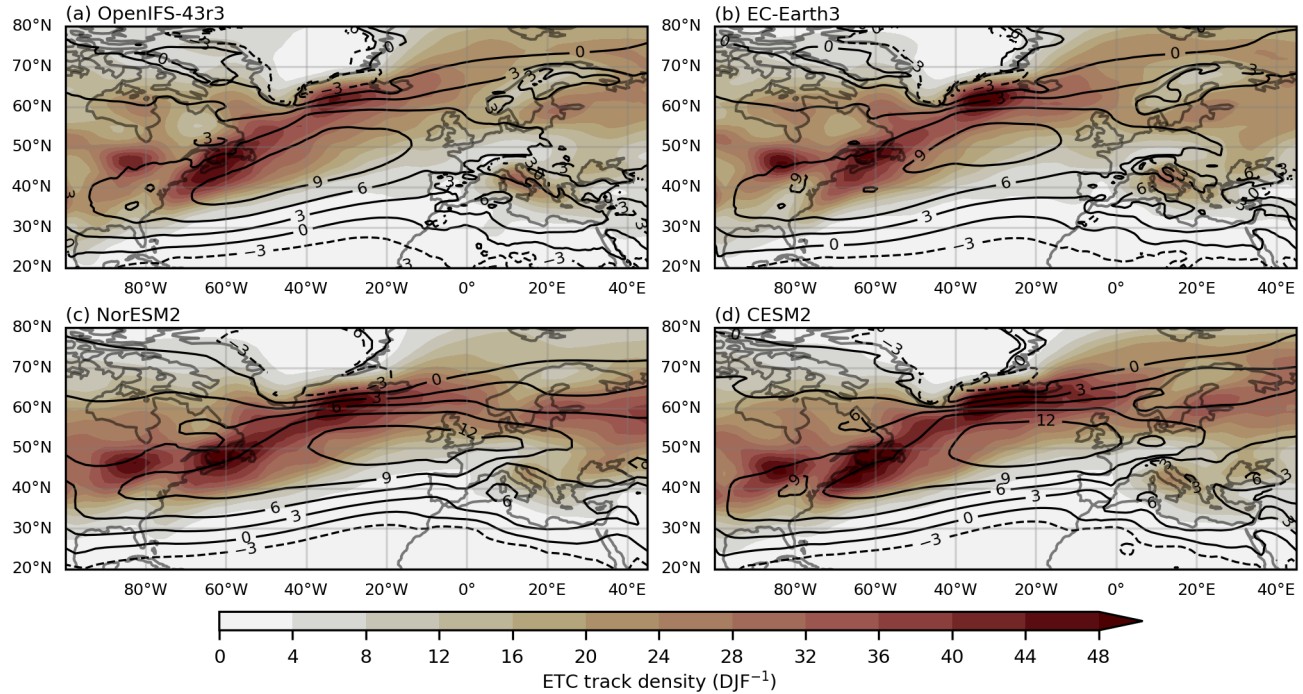

**Figure 2.** The 39-year DJF mean of ETC track density (in colour shading), zonal wind speed $u$ at 850 hPa (in black contours, $\mathrm{m\,s^{-1}}$) for (a) OpenIFS-43r3, (b) EC-Earth3, (c) NorEMS2, and (d) CESM2. The ETC track density unit is number of ETCs per $5°$ spherical cap per winter season (DJF).

addition, this NA jet stream maximum is co-located with a maximum in $MFC$, which contributes to the formation of the NA jet stream.

EC-Earth3 (Fig. 3b) closely matches OpenIFS-43r3 in patterns and magnitudes of $u$, $B_y$ and $MFC$. On the other hand, in CESM2 (Fig. 3d), the NA jet stream is a vertically deep feature, with a strong meridional gradient in wind speed and has a maximum speed between $22\ \mathrm{m\,s^{-1}}$ and $24\ \mathrm{m\,s^{-1}}$. The $MFC$, with values between $100\ \mathrm{m^2\,s^{-2}}$ and $125\ \mathrm{m^2\,s^{-2}}$, is considerably stronger compared to OpenIFS-43r3 and EC-Earth3. This indicates that the eddy-mean flow interaction contributes substantially to the jet structure in CESM2. Meanwhile, the area of strong $B_y$ below the jet maximum is vertically thicker compared to OpenIFS-43r3 and EC-Earth3, which contributes to stronger jet maximum.

Lastly, NorESM2 (Fig. 3c) shares the vertically deep structure and a similar maximum NA jet speed as the CESM2 Baseline simulation. The $B_y$ shows small differences to CESM2, mainly a thicker layer exceeding $12 \cdot 10^{-4}\ \mathrm{s^{-1}}$ at the latitude of the NA jet. Most notably, NorESM2 has the highest values in $MFC$ between $125\ \mathrm{m^2\,s^{-2}}$ and $150\ \mathrm{m^2\,s^{-2}}$). Of all four models, the jet stream in NorESM2 has the highest contribution from eddy momentum flux convergence.

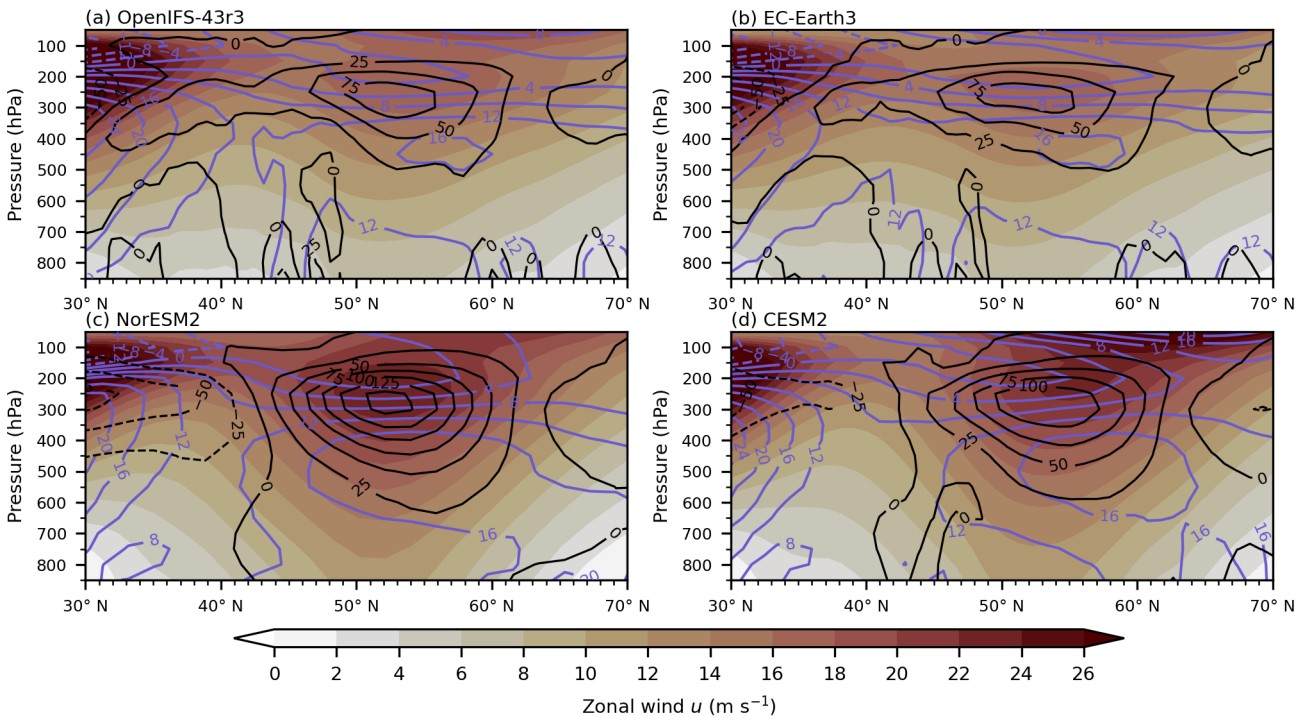

**Figure 3.** Zonal mean ($15° \text{W} - 35° \text{E}$) of the 39-year DJF mean of zonal wind speed $u$ (in colour shading, $\text{m s}^{-1}$), eddy momentum flux convergence $MFC$ (in black contours, $\text{m}^2 \text{s}^{-2}$), and baroclinicity $B_y$ (in blue contours, $10^{-4} \text{s}^{-1}$) for (a) OpenIFS-43r3, (b) EC-Earth3, (c) NorEMS2, and (d) CESM2.

### 3.3 The ETC properties

Looking at the properties of ETCs affecting Europe (defined as spending at least 48 hours within $30° \text{N} - 70° \text{N}$, $15° \text{W} - 35°$ E), Figure 4 provides insight into the ETC count, track duration, mean speed, maximum intensity, the genesis latitude, and latitudinal displacement. All models agree well on the distribution's overall shape and the range of values across all six ETC quantities. However, multiple statistically significant key differences in the mean values are addressed in the following. The detailed differences in the mean values are reported in Table A1 in the Appendix.

The most apparent differences are present in the ETC count. NorESM2 has a significantly lower number of ETCs affecting Europe (1550 total, 39.7 per winter season) than the 3 other models, while CESM2 has the highest number (1814 total, 46.5 per winter season). The mean values for OpenIFS-43r3 (1694 total, 43.4 per winter season) and EC-Earth3 (1720 total, 44.1 per winter season) are not statistically different from each other. Referring back to the track densities in Figure 2, the higher number of ETCs in CESM2 is the origin of the higher ETC track density compared to OpenIFS-43r3 and EC-Earth3. However, regardless of the lower ETC count in NorESM2, NorESM2 presents a higher ETC track density compared to OpenIFS-43r3

and EC-Earth3. This results from the combination of low mean speed and long lifetime of ETCs in NorESM2, which amplifies the ETC track density in the storm track.

The mean lifetime of ETCs is similar in EC-Earth3 (120.6 h) and OpenIFS-43r3 (124.9 h), while it is significantly higher in CESM2 (132.5 h), and NorESM2 (148.8 h). Next, in NorESM2, the mean speed of ETCs with a value of $11.0 \, \mathrm{m \, s^{-1}}$ is significantly slower than in the other three models. Meanwhile, EC-Earth3 has the fastest ETCs on average ($11.9 \, \mathrm{m \, s^{-1}}$). There is no statistical difference in mean speed between CESM2 ($11.3 \, \mathrm{m \, s^{-1}}$) and OpenIFS-43r3 ($11.2 \, \mathrm{m \, s^{-1}}$).

Furthermore, the mean maximum intensity is very similar in OpenIFS-43r3 (33.3 hPa), CESM2 (32.8 hPa), and NorESM2 (32.5 hPa) with no significant differences. EC-Earth3 is the exception, with a significantly higher mean maximum intensity of 35.3 hPa. Moreover, out of the four models, CESM2 has the most equatorward mean latitude of genesis of ETC affecting Europe (39.4° N) and EC-Earth3 the most poleward genesis location (41.8° N). These models differ significantly from OpenIFS-43r3 (40.1° N) and NorESM2 (40.6° N), the difference between the latter two models being insignificant. Lastly, latitudinal displacement is significantly different in all models, in order from least to most poleward: EC-Earth3 (12.8° N), OpenIFS-43r3 (14.0° N), CESM2 (15.1° N), NorESM2 (15.8° N).

## 4 The jet response to climate change

### 4.1 The total Future response of the jet

The response of the upper-level jet stream to changed SSTs and SIC in the SSP5-8.5 scenario is shown in Figure 5. All models show an apparent deceleration on the poleward side of the subtropical jet stream over North Africa. This is due to an upward shift of the subtropical jet maximum (Figure 6), driven by higher SSTs in the tropics and extratropics which leads to a warmer and deeper troposphere. Another common feature across all models is the reduced 250-hPa zonal wind speed on the poleward side of the NA jet stream southeast of Greenland.

The most notable and distinguishing aspects across the models are the changes in wind speed in the NA jet stream at 250 hPa (Fig. 5). In particular, OpenIFS-43r3 only shows a significant increase on the equatorward side of the jet located over the central Atlantic (Fig. 5a). The remaining three models form a group where the jet speed increases on the jet's southern side and in the jet exit region over Europe. To gain more insight into the disagreement between OpenIFS-43r3 and the remaining three models, Figure 6 depicts the zonal mean cross-section over Europe (15° W – 35° E), where the responses differ the most. Evidently, the non-existent response in the NA jet exit over Europe in OpenIFS-43r3 is also visible in the cross-section (Fig. 6a). The only statistically significant features are a deceleration of the polar vortex (60° N – 70° N, 150 – 50 hPa) and an upward shift of the subtropical jet stream. In contrast, EC-Earth3 responds with a barotropic increase of the NA jet stream between $850 \, \mathrm{hPa} - 350 \, \mathrm{hPa}$ and 50° N – 60° N (Fig. 6b). Around the jet maximum at 200 hPa, there is a strong vertical shear in $\Delta FT$, suggesting an additional baroclinic contribution. A similar response in the SSP 5-8.5 scenario to EC-Earth3 is found in NorESM2 (Fig. 6c) and CESM2 (Fig. 6d). However, the maximum increase in zonal wind speed $u$ is located equatorward of the NA jet maximum in the respective Baseline simulation. Furthermore, in NorESM2 and CESM2, the response increases in magnitude with height throughout the troposphere ($850 \, \mathrm{hPa} - 200 \, \mathrm{hPa}$), indicating a more baroclincally-driven response.

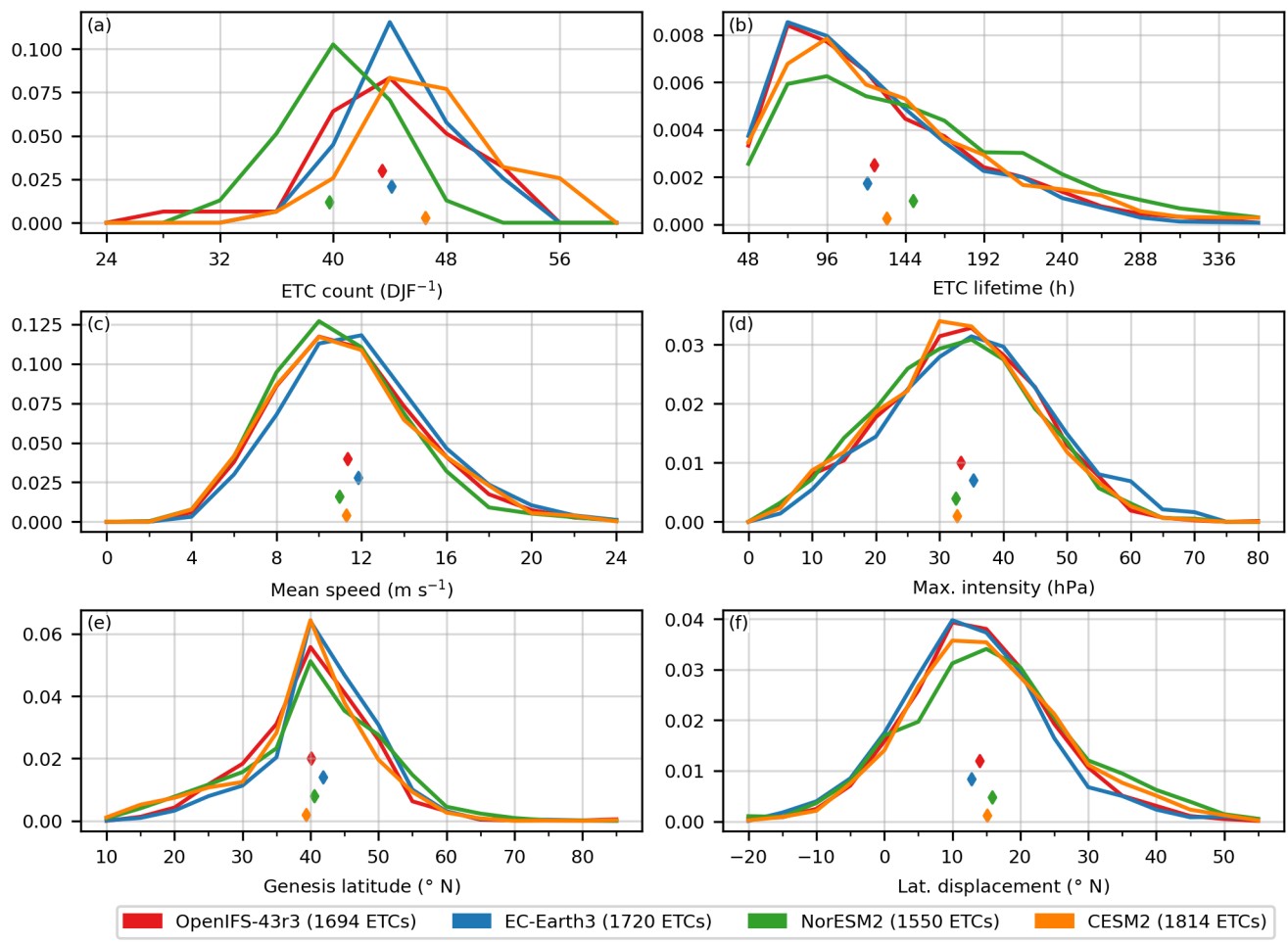

**Figure 4.** The distribution of six ETC quantities and their mean values shown in diamonds for the OpenIFS-43r3 (red, top diamond), EC-Earth3 (blue, second diamond from the top), NorESM2 (green, third diamond from the top), and CESM2 (purple, lowest diamond). Given in brackets in the legend is the total number of ETCs during 39 DJF periods. The six ETC quantities include ETC count per winter season DJF (a), ETC lifetime (b), mean speed (c), maximum intensity (d), genesis latitude (e), and latitudinal displacement (f). The y-axis gives the probability density. Note that the y-axis are different between panels.

## 4.2 The contribution of SST and SIC

A key goal of the present paper is to investigate the contributions of SST and SIC changes to the combined climate response. The SST response, $\Delta SST$, and SIC response, $\Delta SIC$, of the 250 hPa zonal wind are presented in Figure 7. EC-Earth3, NorESM2, and CESM2 show very similar structures in both contributions. $\Delta SST$ (Fig. 7c,e,f) closely resembles the Future response (Fig. 5b,c,d), with an increase on the southern side and in the eastern exit region of the NA jet stream. Hence, the future response is largely dominated by changes in SSTs. The $\Delta SST$ (Fig. 7d,f,h) spatial pattern exhibits a tripole structure.

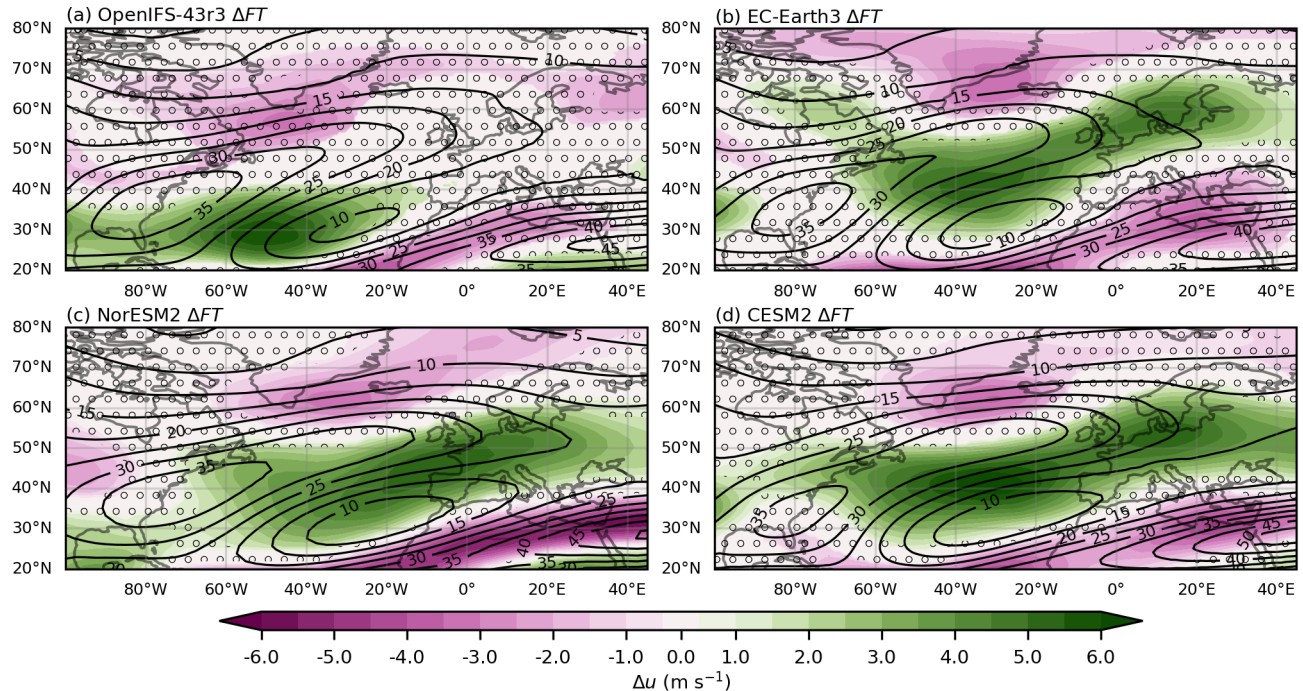

**Figure 5.** Future response $\Delta FT$ in DJF mean of zonal wind speed $u$ at 250 hPa (in colour shading), the Baseline climatology at 250 hPa (in black contours) for (a) OpenIFS-43r3, (b) EC-Earth3, (c) NorEMS2, and (d) CESM2. Shading indicates as consistent sign according to CDR testing and stippling indicates statistically insignificant changes (FDR corrected t-test).

There is a deceleration south of Greenland and around 20° N – 30° N in the Northern Atlantic, and an increase in zonal wind
speed at 40° N – 50° N. The exact geographical location and extent of the tripole structure vary between EC-Earth3, NorESM2, and CESM2. Generally, the increases and decreases in $\Delta SST$ and $\Delta SIC$ are colocated, resulting in an amplifying effect in the Future response.

OpenIFS-43r3 is the outlier in $\Delta SST$ and $\Delta SIC$ regarding the NA jet stream at 250 hPa. The response to SST (Figure 7a) has broadly the same spatial pattern as in the other models, but it is lower in magnitude. The maximum increase is between 4.0
m s$^{-1}$ and 4.5 m s$^{-1}$, while it exceeds 6.0 m s$^{-1}$ in the other models. However, the key difference setting OpenIFS-43r3 apart from the other models is identifiable in the SIC response (Fig. 7b). There is an absence of a tripole structure and a consistent decrease (determined using CDR) relative to the Baseline simulation in the wind speed in the NA jet exit region over Europe. The SST-related increase and SIC-related decrease over Europe compensate each other, resulting in the lack of the response in $\Delta FT$ (Fig. 5a).

The $\Delta SST$ of zonal wind speed cross-section from the OpenIFS-43r3 simulations are shown in Figure 8a. This reaffirms the compensating contributions of $\Delta SST$ and $\Delta SIC$. Due to increased SSTs, the subtropical jet shifts upwards and the NA jet stream increases in speed between 300 hPa and 100 hPa. Meanwhile $\Delta SIC$ shows a reduction in the NA jet speed throughout

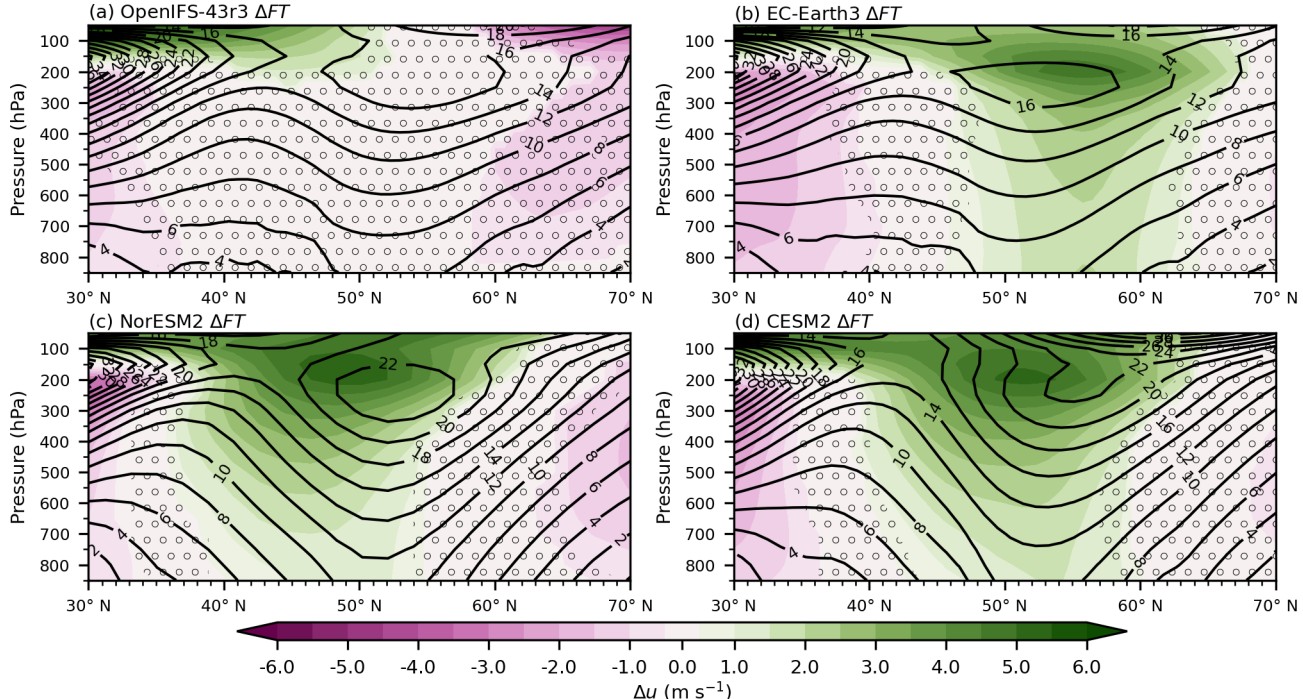

**Figure 6.** Future response $\Delta FT$ of zonal mean (15° W – 35° E) of the 39-year DJF mean of zonal wind speed $u$ (in colour shading), the Baseline climatology (in black contours) for (a) OpenIFS-43r3, (b) EC-Earth3, (c) NorEMS2, and (d) CESM2. Shading indicates as consistent sign according to CDR testing and stippling indicates statistically insignificant changes (FDR corrected t-test).

the troposphere (Fig. 8b). The change in SIC is also the origin of the deceleration of the polar vortex in the Future response (Fig. 8a).

EC-Earth3, NorESM2 and CESM2 present structurally similar $\Delta SST$ for the cross-section of zonal wind speed. The SST responses (Fig. 8c,e,g) show an intensification of the NA jet stream equatorward of jet core in the Baseline simulation. All three models exhibit a positive response in the zonal wind speed through the vertical column, with the strongest response at upper levels (300 hPa – 100 hPa). However, in CESM2 the response is stronger compared to $\Delta SST$ in NorESM2 and EC-Earth. For NorESM2 and CESM2, $\Delta SIC$ (Fig. 8f,h) is characterised by a reduction on the poleward side and an increase on the

equatorward side. Lastly, EC-Earth3 shows no changes to the $u$ cross-section in response to the changes in SIC (Fig. 8d).

## 5   The mechanisms of the jet response

### 5.1   The mechanisms for SST response

A feature common to all models is the impact of SSTs on baroclinicity driving the change of the subtropical jet (Fig. 9a,c,e,g, 30° N – 35° N). The baroclinicity between 850 hPa and 300 hPa decreases, while an increase is visible from 300 hPa to 100

hPa. This corresponds well with the upward shift of the subtropical jet described in Section 4.2. The origin of the increased baroclinicity above 300 hPa is an increased upper-level warming in the tropics (not shown) and an upper-level cooling in the potential temperature in the subtropics/midlatitudes (Fig. 10a,c,e,g, 30° N – 35° N, 200 hPa – 100 hPa). The deepening of the troposphere due to warming SSTs and the subsequent lifting of the thermal tropopause explains the cooling. The tropospheric deepening, and subsequent increase in jet height, also explains the increase in momentum flux convergence located above the
maxima in the respective Baseline simulation for all models (Fig. 11a,c,e,g, 40° N – 60° N, 300 hPa – 100 hPa).

In OpenIFS-43r3 (Fig. 9a), the baroclinicity associated with the NA jet exit (45° N – 65° N) increases at upper levels (400 hPa – 200 hPa); this results in the shallow response of the zonal wind speed seen in Figure 8a. The increase in baroclinicity is caused by a larger increase in potential temperature at lower latitudes (30° N – 50° N) compared to higher latitudes (50° N – 70° N). This meridional difference in potential temperature increase is the strongest between 400 hPa – 300 hPa (Fig. 10a).
$MFC$ increases at the location of the zonal wind speed increase (45° N – 65° N, 200 hPa – 100 hPa). Like the zonal wind speed response, OpenIFS-43r3 shows the weakest response in $MFC$ among all models.

EC-Earth3 shows a very similar SST response to OpenIFS-43r3 with two distinct differences (Fig. 9c). First, the upper-level baroclinicity increase (45° N – 65° N, 400 hPa – 200 hPa) is stronger. This is the consequence of a stronger upper-level warming in EC-Earth3 (Fig. 10c, 30° N – 40° N, 400 hPa – 200 hPa). Furthermore, the enhanced baroclinicity response
explains the stronger increase in the upper-level part of the NA jet stream exit in EC-Earth3 compared to OpenIFS-43r3, as shown in Section 4.2. Second in EC-Earth3, there is a weak increase in baroclinicity from 40° N to 50° N and between 700 hPa and 400 hPa, which results from a more pronounced warming at the subtropical middle troposphere (30° N – 40° N, 400 hPa – 700 hPa). However, this increase in baroclinicity does not explain the increase in zonal wind speed at lower levels (500 hPa – 850 hPa) in Figure 8c, as the maximum is located between 49° N – 54° N. This requires the momentum flux convergence (Fig.
11c), which shows an increase at the lower levels in EC-Earth3 (47° N – 53° N, 500 hPa – 850 hPa). An increase in $MFC$ leads to a barotropic increase in zonal wind speed, present in Figure 8c. Furthermore, there is a decrease in $MFC$ between 60° N and 70° N at upper levels (400 hPa – 200 hPa), which is co-located with an increase in baroclinicity. These effects are compensating, resulting in a reduced increase in zonal wind speed co-located with a $MFC$ decrease.

The $\Delta SST$ of baroclinicity in NorESM2 (Fig. 9e) is structurally similar to EC-Earth3. The strongest increases in baroclin-
icity are found at 400 hPa to 200 hPa and from 40° N to 60° N. Additionally, NorESM2 also features a weak but significant increase in baroclinicity at lower levels (500 hPa – 850 hPa, 35° N – 50° N), which is more pronounced relative to EC-Earth3. Contrary to EC-Earth3, the origin of the baroclinicity response in NorESM2 is slightly different. Figure 10e shows that the upper-level warming between 30° N and 45° N is weaker in NorESM2 compared to EC-Earth3. Likewise, the upper-level potential temperature increase between 45° N to 70° N is weaker, resulting in a comparable baroclinicity response. NorESM2,
like the other models, shows the strongest increase in $MFC$ co-located with the zonal wind speed increase (Fig. 11e). More-over, like EC-Earth3, NorESM2 exhibits the compensating effect of baroclinicity and $MFC$, found at 60° N to 70° N and 400 hPa to 200 hPa, resulting in a non-significant change in zonal wind speed (Fig. 8e).

Lastly, CESM2 shows a baroclinicity response structurally similar to NorESM2, but with an amplified magnitude (Fig. 9g). This results from weaker warming of the atmospheric column at higher latitudes (55° N – 70° N) in CESM2 compared to

NorESM2, while the warming at lower latitudes (30° N – 55° N) is comparable (Fig. 10g). Unlike the other models, the change in baroclinicity is largest south of maximum in the Baseline simulation. Together with the change in $MFC$, this increases the jet speed located southward of the maximum in the Baseline simulation. Furthermore, $MFC$ contributes a weak amplification of the zonal wind speed at lower levels (850 hPa – 600 hPa, 52° N – 58° N) in Figure 11g.

### 5.2 The mechanisms for SIC response

OpenIFS-43r3 shows the geographically most extensive reduction in baroclinicity across all models (Fig. 9b), with large decreases spanning from 47° N to 70° N and from 850 hPa to 300 hPa. The baroclinicity response originates from deep vertical warming at 60° N to 70° N (Fig. 10b) with the strongest warming located at the surface. This warming is directly induced by sea ice removal, exposing the sea surface to the atmosphere. A secondary effect is a cooling at lower latitudes (35° N – 50° N), which has its maximum at 500 hPa to 300 hPa. This dipole structure has been identified in previous studies (Screen et al., 2022; Labe et al., 2020). Furthermore, $MFC$ shows a non-significant reduction colocated with its Baseline maximum (Fig. 11b). However, the change in baroclinicity is the primary driver of the zonal wind speed response to sea ice loss and $MFC$ plays a secondary role.

The response of the zonal wind speed to changes in sea ice in EC-Earth3 does not show any significant changes. This is the result of opposing weak effects of $MFC$ and $B_y$. EC-Earth3's baroclinicity shows the weakest $\Delta SIC$ of all models (Fig. 9d) which is due to the weaker vertical and broader horizontal extent of warming due to sea ice loss. Relevant to the NA jet stream exit is the reduction in $B_y$ is found at 50° N to 60° N and 850 hPa to 600 hPa, above which a consistent minor increase in $MFC$ is present. A similar structure is found at 65° N to 70° N, where a $MFC$ decrease is associated with an increase in $B_y$. $B_y$ and $MFC$ are acting in opposite directions in the jet exit region in EC-Earth3.

The dominating factor in NorESM2 is $MFC$ (Fig. 11f). There is a dipole with an increase on the equatorward side and a decrease on the poleward side of the Baseline maximum. This results in the equatorward shift of the NA jet stream exit in NorESM2. Supporting the equatorward shift in zonal wind speed is a decrease in baroclinicity on the poleward side (60° N – 70° N, 850 hPa – 700 hPa) and an increase on the equatorward side (40° N – 50° N, 500 hPa – 300 hPa), shown in Figure 9f. The cause of the reduction in baroclinicity is shallow warming at high latitudes (60° N – 70° N, 850 hPa – 700 hPa) resulting directly from the removal of sea ice (Fig. 10f). Additionally, the increase in $B_y$ results from a warming at 30° N to 40° N and a cooling at 50° N to 65° N.

CESM2 responds to the decreased SIC with a vertically extensive reduction in $B_y$ reaching from 850 hPa to 400 hPa located between 53° N and 70° N (Fig. 9h). This reduction originates similarly to OpenIFS-43r3 from deep high-latitude vertical warming (60° N – 70° N, 850 hPa – 400 hPa) and an adjacent cooling equatorward of the warming (Fig. 10h). The $MFC$ presents a dipole structure, which is not associated with a corresponding dipole in the zonal wind speed, indicating the baroclinicity change is the dominating factor for the SIC response in CESM2.

## 6   The extra-tropical cyclone response

### 6.1   The Future response of ETC track density

Figure 12 shows the Future response of the ETC densities for all four models. In OpenIFS-43r3, the majority of the ETC track density changes compared to the Baseline simulation are non-significant. Significant reductions are, however, present in the location of the storm track maxima over the Eastern US and Western Atlantic. Furthermore, Europe (30° N – 70° N, 15° W – 35° E) sees a non-significant reduction in ETC track density.

Next, in EC-Earth3, $\Delta FT$ shows a clearer signal (Fig. 12b). There is an increase in ETC track density exceeding 4 ETCs per 5° spherical cap per winter season downstream of the Baseline storm track. This increase is tied to the eastward extension of the NA jet stream. Specifically, the ETC track density is increased northward of the zonal wind speed increase. Additionally, there is a significant decrease east of Greenland and over the European continent.

NorESM2 is set apart by a significant decrease in ETC track density in the storm track in the Future simulation (Fig. 12c). The maximum decrease is found around Iceland with a magnitude > 12 ETCs per 5° spherical cap per winter season. Furthermore, the ETC track density in the Mediterranean is reduced by 2 ETCs per 5° spherical cap per winter season. Similarly to EC-Earth3, the increase in ETC track density over Northern Europe is tied to the increase in the NA jet stream.

CESM2 presents an increase downstream of the Baseline storm tracks (Fig. 12d). As with other models, this is co-located with the increase the NA jet stream in Future simulation. Also, CESM2 show its strongest reduction in ETC track density eastward of Greenland, and the Western Atlantic. Over the Mediterranean, there is a relatively weaker, nonetheless significant, decrease in the ETC track density.

### 6.2   The SST and SIC contributions

Due to the sparser nature and higher variability of the ETC dataset compared to, for example, the zonal wind speed, the contribution of SSTs and SIC do not sum up to the Future response, resulting in a larger residual. Nonetheless, it is useful to look at the effects of changed SST and SIC individually.

OpenIFS-43r3 shows a reduction of ETC track density over the Eastern US and Western Atlantic as a response to increased SSTs (Fig. 13a). There is an increase in ETC in the Eastern Atlantic (48° N – 60° N, 40° W – 5° W), extending the storm track to the east. This eastward extension is located north of the zonal wind increase (Fig. 7a). Meanwhile, in the SIC response (Fig. 13b), a clear feature is a dipole southeast of Greenland, effectively leading the southward shift. In the Mediterranean, $\Delta SST$ and $\Delta SIC$ are opposed to each other, where $\Delta SST$ showing a decrease in ETC track density and $\Delta SIC$ showing an increase.

EC-Earth3's $\Delta SST$ is characterised by a shift of the storm track towards the southeast (Fig. 13c). The strongest decrease takes place between Greenland and Iceland. Moreover, there is a significant decrease in the Mediterranean area and the Arctic Ocean. Meanwhile, there is an extensive increase in ETC track density over the Eastern Atlantic Ocean (40° N – 58° N, 40° W – 5° E). The SIC response presents a similar dipole (Fig. 13d), with a reduction southeast of Greenland and an increase over the central Atlantic (40° N – 58° N, 50° W – 20° W). The compounding effects of $\Delta SST$ and $\Delta SIC$ explain the clearer increase over Northern Europe and decrease over Central Europe in the Future response in EC-Earth3. Both $\Delta SST$ and $\Delta SIC$

show a strong connection with the jet stream changes (Fig. 7c,d). The widespread increases in ETC track density are located northward of the increase in jet speed.

For NorESM2, $\Delta SST$ (Fig. 13e) strongly contributes to the reduction in ETC track density in the Future response. This reduction is primarily located in the storm track from the Eastern US along the Western Atlantic towards Greenland. Additionally, the change in SST is the origin of the decrease in the Mediterranean. While NorESM2 shows a similarly strong increase in zonal wind speed in $\Delta SST$ as EC-Earth3, NorESM2 shows a weaker positive response in the ETC track density compared to EC-Earth3. Meanwhile, the change in SIC (Fig. 13f) is responsible for the reduction located over Iceland and the Arctic Ocean. SSTs and SIC contribute to an increase over Northern Europe and the British Isles, with $\Delta SIC$ having a larger magnitude (8 to 9 ETCs per 5° spherical cap per winter season).

In CESM2, similar to NorESM2, the changed SSTs reduce the ETC track density along the Baseline simulation storm track and the Mediterranean (Fig. 13g). However, the reduction in the Arctic Ocean due to SST changes is more pronounced in CESM2. The increases in ETC track density in the North Atlantic are statistically insignificant, and like the other models, they are located northward of the increases in the zonal wind. The SIC response has the largest impact in the Arctic Ocean (Fig. 13h, 60° N – 80° N, 25° W – 20° E). The Future response of CESM2 shows a clearer signal compared to the individual contributions of SST and SIC. However, the individual contributions have similar spatial patterns, indicating that the Future response in CESM2 is not dominated by either SST or SIC.

## 6.3    The impacts on ETC properties

Figure 14 presents the changes in the mean value of ETC count, ETC lifetime, mean speed, maximum intensity, genesis latitude, and latitudinal displacement for ETC track affecting Europe (at least 48 hours within 30° N – 70° N, 15° W – 35° E). Similar to the track densities, $\Delta SIC$ and $\Delta SST$ do not necessarily add linearly to exactly equal the Future response. However, they provide insight into the dominating contribution to the Future response and help to compare the responses across models. The diamonds in Figure 14 represent the mean value of the ETC properties in the Baseline simulation, which has been previously discussed in Section 3.3 (Figure 4). The inclusion of the Baseline simulation sets the responses of each model in the context of model differences. Considering ETC properties, the majority of Future, SIC and SST responses are smaller than the differences between the individual models. In particular, ETC count and genesis latitude are properties which show responses of a magnitude similar to the model differences.

The ETC count per winter season DJF shows a decrease in the Future response originating from $\Delta SST$ (Fig. 14a) across models, with the change being statistically significant in NorESM2 ($\Delta FT$, -2.3 DJF$^{-1}$) and CESM2 ($\Delta FT$, -4.5 DJF$^{-1}$). Models disagree on the effect of changed SIC on ETC count. However, OpenIFS-43r3 has the largest change amongst models by +2.4 per winter season DJF, which is corresponds to the increase in ETC track density found in the Mediterranean (Fig. 13b).

The SIC and SST response of ETC lifetime contribute similarly in magnitude to the Future response (Fig. 14b). EC-Earth3 and CESM2 present an increase in lifetime for $\Delta SST$ and $\Delta SIC$, which results in a significant increase in the Future response (EC-Earth3: +5.0 h, CESM2: +7.5 h). Meanwhile, OpenIFS-43r3 shows a non-significant decrease in ETC lifetime for $\Delta SST$

and an increase for $\Delta SIC$, which compensate each other in the Future response. There are only minimal, non-significant changes to the ETC lifetime in NorESM2. The mean speed of ETCs does not show any significant changes (Fig. 14c), besides a deceleration in EC-Earth3's Future response (-0.27 m s$^{-1}$). Additionally, the models disagree on the sign of the change for the Future, SST and SIC response.

For the mean maximum intensity (quantified by the magnitude of the pressure anomaly), models agree on an increase in response to the SST changes, indicating a deepening of ETCs, and a decrease in response to the SIC changes, corresponding to a reduction of the pressure minimum (Fig. 14d). However, many of these changes are not significant. NorESM2 shows a significant decrease due to SIC changes (-1.8 hPa) and a minimal non-significant change due to $\Delta SST$, which results in a reduced maximum intensity on average in the Future response. Next, while EC-Earth3 presents a significant increase in maximum intensity for $\Delta SST$, it shows a non-significant decrease in $\Delta FT$, and $\Delta SIC$.

The Future response of the genesis latitude is dominated by the change in SSTs with a significant increase across all models (Fig. 14e). Averaged across models, the genesis latitude increases by +1.1° N in $\Delta FT$ and +1.3° N in $\Delta SST$. The SIC responses are non-significant across models. Figure 14f shows the latitudinal displacement of ETCs which is significantly reduced in OpenIFS-43r3, EC-Earth3, and NorESM2 for the Future response. For NorESM2, the change originates from the SST change and the interaction of the SST and SIC changes. Meanwhile, $\Delta SIC$ dominates for OpenIFS-43r3. The SST responses are not significant in any of the four models. Combining the changes in the genesis latitude and latitudinal displacement ETCs are propagating further north in the SST change only simulation. However, in the Future climate scenario ETCs travel more zonally in OpenIFS-43r3, EC-Earth3, NorESM2, which indicates important interactions of SST and SIC changes.

## 7 Discussion

Multiple aspects of the results require a more detailed discussion. Foremost, it is important to consider the robustness of the results obtained here given that a previous study (Peings et al., 2021) suggested that longer simulations, or larger ensembles, are required to isolate consistent atmospheric responses to sea ice loss. The atmospheric response due to sea ice loss found in this study is weak in the midlatitudes. This may be partially caused by the models underestimating the responses due to a too-weak eddy feedback, as shown previously by Smith et al. (2022) and Screen et al. (2022). Nonetheless, consistent signals in the sign of the response are identified using the CDR leading to confidence in the sign of the response if not the magnitude of the response. In comparison to previous studies, the sea ice loss forcing applied here is stronger, making it easier to distinguish the response from internal variability. Furthermore, the SIC response combined with the responses due to changing SST are coherent with the Future responses, where both SST and SIC are changed. While the results shouldn't be overinterpreted, they provide evidence for the physical mechanisms contributing to the future change to the North Atlantic jet stream and storm track in addition to the largely descriptive results of how the jet and storm track change.

The Future winter North Atlantic jet stream (identified at 250 hPa, Fig. 5) intensifying on its southern flank and extending further into Europe in EC-Earth3, CESM2, NorESM2 is consistent with findings across CMIP3, CMIP5, CMIP6 by Harvey et al. (2020), albeit for weaker warming scenario SSP 2-4.5 compared to this study (SSP 5-8.5). The remaining Future response

of OpenIFS-43r3 is a scientifically important outlier, particularly as it is the same atmospheric model component used in EC-Earth3, albeit a much newer version. The SST response is structurally consistent across all models (Fig. 7). The sea-ice loss induced reduction of jet speed at 250 hPa over Europe in OpenIFS-43r3, CESM2, and NorESM2 is present in the PAMIP simulation (Ye et al., 2023). However, the spatial overlap of the SST and SIC response in OpenIFS-43r3 is the important difference to the other models, which explains the lack of an extension of jet stream over Europe. Using greenhouse gas and sea ice loss forcing, Oudar et al. (2020) have shown a similar compensating effect on zonal wind speed at 850 hPa over Europe.

There are striking differences in the vertical extent of the sea-ice loss induced warming at high latitudes across models (Fig. 10). The vertical extent of high latitude warming has been shown to be important for mid-latitude atmospheric circulation. For example, the vertical extent of high latitude warming has been investigated as a contributor to the Eurasian cooling pattern in future climate projections (He et al., 2020; Labe et al., 2020). Moreover, idealised simulations suggest an enhanced reduction in the zonal wind speed with high latitude warming aloft compared to the same magnitude of warming near the surface (Kim et al., 2021). Results by Xu et al. (2023) indicate that sea-ice loss in the Barents-Kara sea contributes to a stronger high latitude warming aloft, which in turn impact mid-latitude circulation. Similarly to (Labe et al., 2020), this study shows enhanced upper-tropospheric cooling in the mid-latitudes with higher vertical extent of warming at high latitudes in OpenIFS-43r3 and in weaker form in CESM2 (Fig. 10b,h). This is associated with a decrease in baroclinicity in the mid-troposphere (Fig. 9b,h) which impacts the jet speed over Europe (Fig. 8b,h). In contrast, EC-Earth3 shows a moderate vertical extent of warming without an associated upper-tropospheric cooling in the mid-latitudes, which results in weak changes in baroclinicity and no significant impact on zonal wind speed. Lastly, despite a shallow vertical extent in high latitude warming, NorESM2 shows a weak deceleration of zonal wind speed, which is explained by the changes driven by changes in the momentum flux convergence rather than in baroclinicity.

The Future response of the storm tracks found in this study closely resemble results from CMIP6 (Priestley and Catto, 2022), where it was shown that under the SSP 5-8.5 scenario the ETC track density increases over the British Isles and Northern Europe, while it decreases in the Mediterranean. Previous studies showed that the North Atlantic storm track shifts south in response to sea-ice loss (Hay et al., 2023; Yu et al., 2023). This study supports these finding. However, this southward shift is model dependent, with EC-Earth3 and NorESM2 showing strong responses while CESM2 and OpenIFS-43r3 are comparatively weak (Fig. 13). In contrast to Yu et al. (2023), $\Delta SST$ and $\Delta SIC$ have contributions of similar magnitude to the Future response of the North Atlantic storm track for each individual model. It is important to note that the storm tracks exhibit high variability, therefore the precise decomposition of the contributions of SST and SIC, including the non-linear interactions between SST and SIC, requires further investigation with statistical methods like principal component analysis.

Furthermore, the sea-ice response results of the ETC properties agree with Hay et al. (2023), who show that ETCs tend to have a longer lifetime and a reduced intensity. Similarly, the number of ETCs reduce in most models (EC-Earth3, CESM2, NorESM2), while it increases in others (OpenIFS-43r3). In agreement with CMIP6 results, the Future response of all four models agree on a reduction in the number of ETCs (Priestley et al., 2023) and a more zonal propagation (Crawford et al., 2023).

## 8  Conclusions

This study presents the responses of the North Atlantic jet stream and storm track to future sea surface temperatures and sea ice cover and quantifies their contributions using the set of atmosphere-only model simulations from the CRiceS project (CRiceS). Furthermore, the potential driving mechanism of the responses were examined by using the baroclinicity and momentum flux convergence. The key findings of this study are as follows:

- The Baseline simulations of the four models agree on the general shape of the North Atlantic jet stream, which is in reasonable agreement with results from ERA5 reanalysis and CMIP6 (Harvey et al., 2020). A key difference is that the zonal wind speed $u$ in the upper troposphere is higher over Europe in NorESM2 and CESM2 compared to EC-Earth3 and OpenIFS-43r3. While the baroclinicity is comparable across models, the difference in the jet strength over Europe is primarily explained by the magnitude of momentum flux convergence in the Baseline simulations, which is larger in NorESM2 and CESM2. The ETC track density is similar across models and the patterns of the ETC track density in the Baseline simulation closely match previous findings based on CMIP6 simulations (Priestley et al., 2020), as well as PAMIP (Yu et al., 2023).

- In the Future response, three out of the four models show an increase in zonal wind speed at 250 hPa on the southeastern side of the Baseline jet core over the eastern North Atlantic and reaching far into Europe, which has been shown in CMIP6 simulations (Harvey et al., 2020). The $\Delta SST$ contribution dominates the Future response by an increase in the upper tropospheric baroclinicity. The momentum flux convergence contributes to an equatorward shift of the NA jet stream in the SIC response.

- The exception is OpenIFS-43r3 with no significant change in zonal wind speed over Europe. The SST response is similar to the other three models, meanwhile, the SIC response shows a decrease in zonal wind speed over Europe, which is driven by a decrease by more than 10 % in baroclinicity at lower levels (850 hPa – 400 hPa). These overlapping opposite responses result in no significant changes to the jet stream over Europe in the Future scenario.

- The ETC track density reduces along the East coast of Northern America and Greenland in the Future scenario. SST and SIC changes both significantly contribute to the reduction. A consistent feature is a significant reduction of ETC track density in the Mediterranean basin, which is driven by the changes in SSTs. The increases in ETC track density over the Eastern North Atlantic, leading to an eastward extension of the storm track, are associated with the extension of the NA jet stream. This eastward extension and reduction in the Mediterranean basin is also present in CMIP projections (Priestley and Catto, 2022). Similar to the jet stream changes, OpenIFS-43r3 shows no significant increases over Europe in the ETC track density in the Future.

- The future changes to the mean values of the ETC-specific properties are superseded by the inter-model differences. For example, mean speed and maximum intensity show no consistent statistically significant changes for two or more models. However, ETC affecting Europe tend to become less frequent, which is linked to the reduced ETC track density

in the Mediterranean. They also originate more poleward in the Future scenario, which is driven by changes in SSTs, and travel more zonally, driven by the interaction of SST and SIC changes.

- The total future response of the North Atlantic at 250 hPa is dominated by the changes in sea surface temperature. On the other hand, the contributions of changes in sea ice cover and sea surface temperature to the total future response of the North Atlantic storm track are of similar magnitude.

This study is based on atmosphere-only simulations with annually repeating sea-ice cover and sea surface temperatures, which limits the direct comparability between the studied simulations and results from reanalysis and CMIP. Furthermore, the changes presented are the mean responses of the jet stream and storm track, leaving the investigation of changes to the variability and extreme events of the jet stream and storm track to future research. Moreover, one aim of the coordinated simulations is to constrain the variability of the climate system. However, the internal atmospheric climate variability is a major source of uncertainty in this study. This affects both the study between the differences between simulations within one model and the comparison between the four different models. In particular, the metrics related to extratropical cyclones are strongly affected by atmospheric internal variability. In addition, motivated by the model-specific physical interpretability, the number of models is limited. Thus, it is difficult to assess whether the disagreement of the OpenIFS-43r3 response is an outlier or a representative of a different part of spread of state-of-the-art models.

This study encourages further investigation into the interannual variability of the baroclinicity and eddy-driven mechanisms of the jet stream and how this connects to the storm track activity. Moreover, as this study investigates the mean changes to extratropical cyclones, further work on extreme cyclones and associated extreme events of wind and precipitation.

Notable differences in the Future responses occur, despite the models having similar Baseline climates, and being forced with the same changes to SSTs and sea-ice cover. Furthermore, although the SST response dominates, the sea ice cover (SIC) response is significant in some areas. For example, in OpenIFS-43r3 the SIC response is large enough to counteract the impact of changing SSTs. Overall, substantial uncertainties remain in how the jet stream and storm track in the Northern Atlantic and Europe will change in the future.

*Code and data availability.* The cyclone tracking algorithm TRACK is available on GitHub (Hodges, 2020). The data and scripts necessary to produce the plots are available on Zenodo (Köhler et al., 2024).

The full CRiceS simulation data set is available as follows:

OpenIFS-43r3: https://a3s.fi/CRiceS_Index/CRiceS_index.html. EC-Earth3: https://crices-task33-output-ecearth.lake.fmi.fi/index.html and https://crices-task33-output-ecearth-ifs-monthly-means.lake.fmi.fi/index.html. NorESM2: At the moment NorESM2 data is available from authors upon request and it will be published to a public archive during the review process. CESM2: https://archive.sigma2.no/pages/public/datasetDetail.jsf?id=10.11582/2024.00018.

The model code is available as follows:

OpenIFS-43r3: Documentation is available at https://confluence.ecmwf.int/display/OIFS. The licence for using the OpenIFS model can be requested from ECMWF user support (openifs-support@ecmwf.int). EC-Earth3: Brief general documentation of EC-Earth3 is provided at

610 https://ec-earth.org/ec-earth/ec-earth3/. The code is available to registered users at https://ec-earth.org/ec-earth/ec-earth-development-portal/. Only employees of institutes that are part of the EC-Earth consortium can obtain an account. NorESM2: Documentation is available at https://www.noresm.org/. The code is available at https://github.com/NorESMhub/NorESM. CESM2: documentation is available at https://escomp.github.io/CESM/versions/cesm2.2/html/. The code is available at: https://github.com/ESCOMP/CESM.

## Appendix A

| ETC count (DJF$^{-1}$) | OpenIFS-43r3 | EC-Earth3 | NorESM2 | CESM2 | Track duration (h) | OpenIFS-43r3 | EC-Earth3 | NorESM2 | CESM2 |
|---|---|---|---|---|---|---|---|---|---|
| OpenIFS-43r3 | (43.436) | -0.667 | **3.692** | **-3.077** | OpenIFS-43r3 | (124.867) | 4.249 | **-23.910** | **-7.620** |
| EC-Earth3 | 0.667 | (44.103) | **4.359** | **-2.410** | EC-Earth3 | -4.249 | (120.617) | **-28.159** | **-11.869** |
| NorESM2 | **-3.692** | **-4.359** | (39.744) | **-6.769** | NorESM2 | **23.910** | **28.159** | (148.777) | **16.291** |
| CESM2 | **3.077** | **2.410** | **6.769** | (46.512) | CESM2 | **7.620** | **11.869** | **-16.291** | (132.486) |
| Mean speed (m s$^{-1}$) | OpenIFS-43r3 | EC-Earth3 | NorESM2 | CESM2 | Max. intensity (hPa) | OpenIFS-43r3 | EC-Earth3 | NorESM2 | CESM2 |
| OpenIFS-43r3 | (11.343) | **-0.509** | **0.370** | 0.071 | OpenIFS-43r3 | (33.291) | **-2.029** | **0.764** | 0.540 |
| EC-Earth3 | **0.509** | (11.852) | **0.879** | **0.580** | EC-Earth3 | **2.029** | (35.320) | **2.793** | **2.569** |
| NorESM2 | **-0.370** | **-0.879** | (10.972) | **-0.299** | NorESM2 | **-0.764** | **-2.793** | (32.527) | -0.224 |
| CESM2 | -0.071 | **-0.580** | **0.299** | (11.271) | CESM2 | -0.540 | **-2.569** | 0.224 | (32.750) |
| Lat. of gen. ($^\circ$ N) | OpenIFS-43r3 | EC-Earth3 | NorESM2 | CESM2 | Lat. displ. ($^\circ$ N) | OpenIFS-43r3 | EC-Earth3 | NorESM2 | CESM2 |
| OpenIFS-43r3 | (40.130) | **-1.702** | **-0.471** | 0.779 | OpenIFS-43r3 | (13.971) | **1.182** | **-1.872** | **-1.149** |
| EC-Earth3 | **1.702** | (41.832) | **1.231** | **2.481** | EC-Earth3 | **-1.182** | (12.789) | **-3.054** | **-2.331** |
| NorESM2 | **0.471** | **-1.231** | (40.601) | **1.250** | NorESM2 | **1.872** | **3.054** | (15.843) | **0.723** |
| CESM2 | -0.779 | **-2.481** | **-1.250** | (39.351) | CESM2 | **1.149** | **2.331** | **-0.723** | (15.120) |

**Table A1.** Additional information to Figure 4. For the six ETC quantities, the mean value for each model is reported on the diagonal (in parentheses) and the differences between the models (row - column) are given in the off-diagonal cells. A Mann-Whitney U test p-value < 0.05 is highlighted in bold font.

*Author contributions.* DK and VS contributed to the design of the study. DK performed the analysis with guidance from VS. DK wrote the publication, supported by VS, PR, TN, and KN. The model simulations have been conducted by DK (OpenIFS-43r3), PR (EC-Earth3), TN (NorESM2), KN (CESM2).

*Competing interests.* The authors declare that they have no conflict of interest.

*Acknowledgements.* This project has received funding from the European Union's Horizon 2020 research and innovation programme under
620 grant agreement No 101003826 via project CRiceS (Climate Relevant interactions and feedbacks: the key role of sea ice and Snow in the

polar and global climate system). This research was also supported by the Academy of Finland (grant no. 338615). We wish to thank Kevin Hodges for providing the cyclone tracking software TRACK. We acknowledge CSC – IT Centre for Science, Finland, for providing generous computational resources.

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

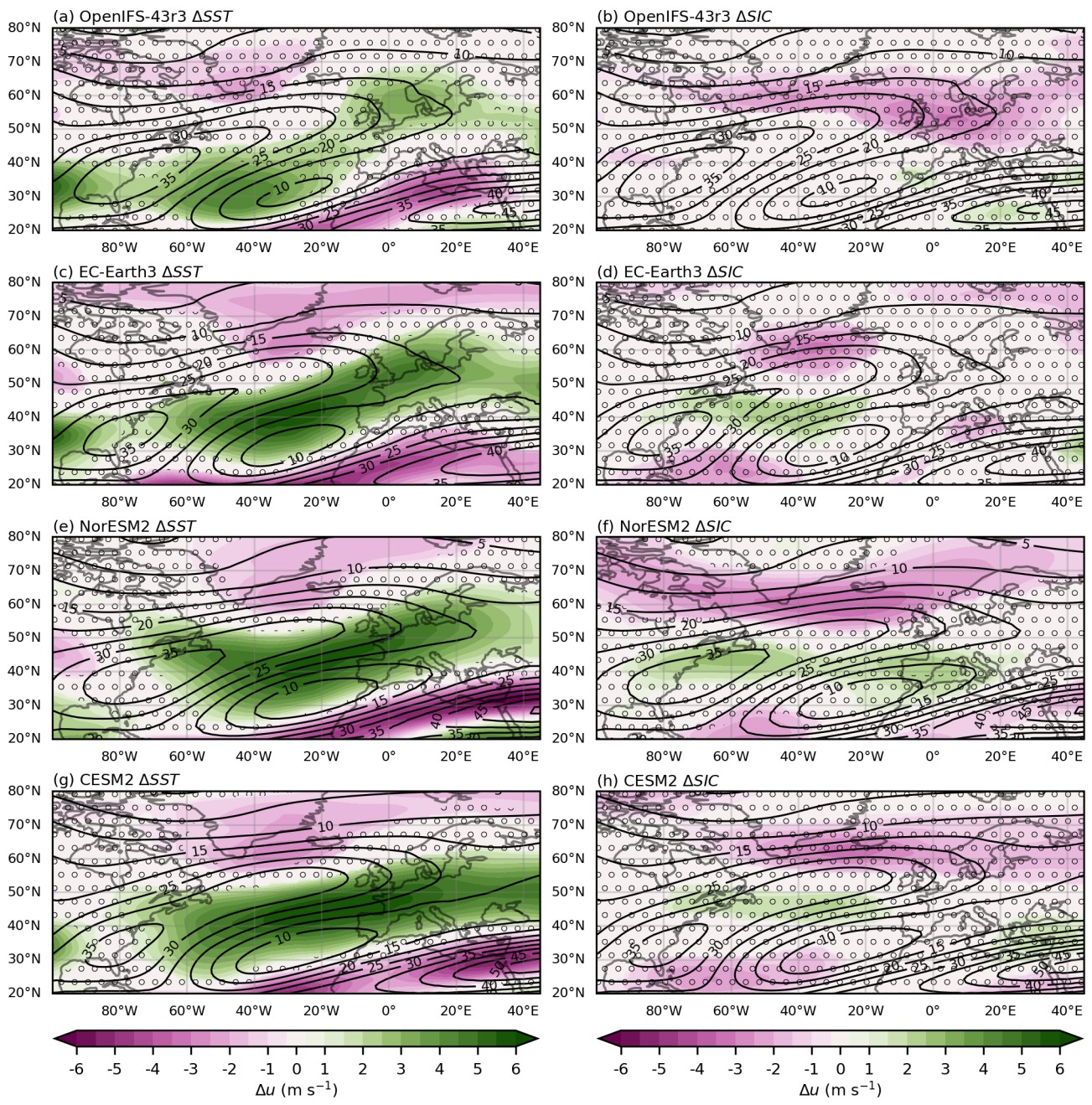

**Figure 7.** SST response $\Delta SST$ (a,c,e,g) and SIC response $\Delta SIC$ (b,d,f,h) of DJF mean of zonal wind speed $u$ at 250 hPa (in colour shading), the Baseline climatology at 250 hPa (in black contours) for (a) & (b) OpenIFS-43r3, (c) & (d) EC-Earth3, (e) & (f) NorEMS2, and (g) & (h) CESM2. Shading indicates as consistent sign according to CDR testing and stippling indicates statistically insignificant changes (FDR corrected t-test).

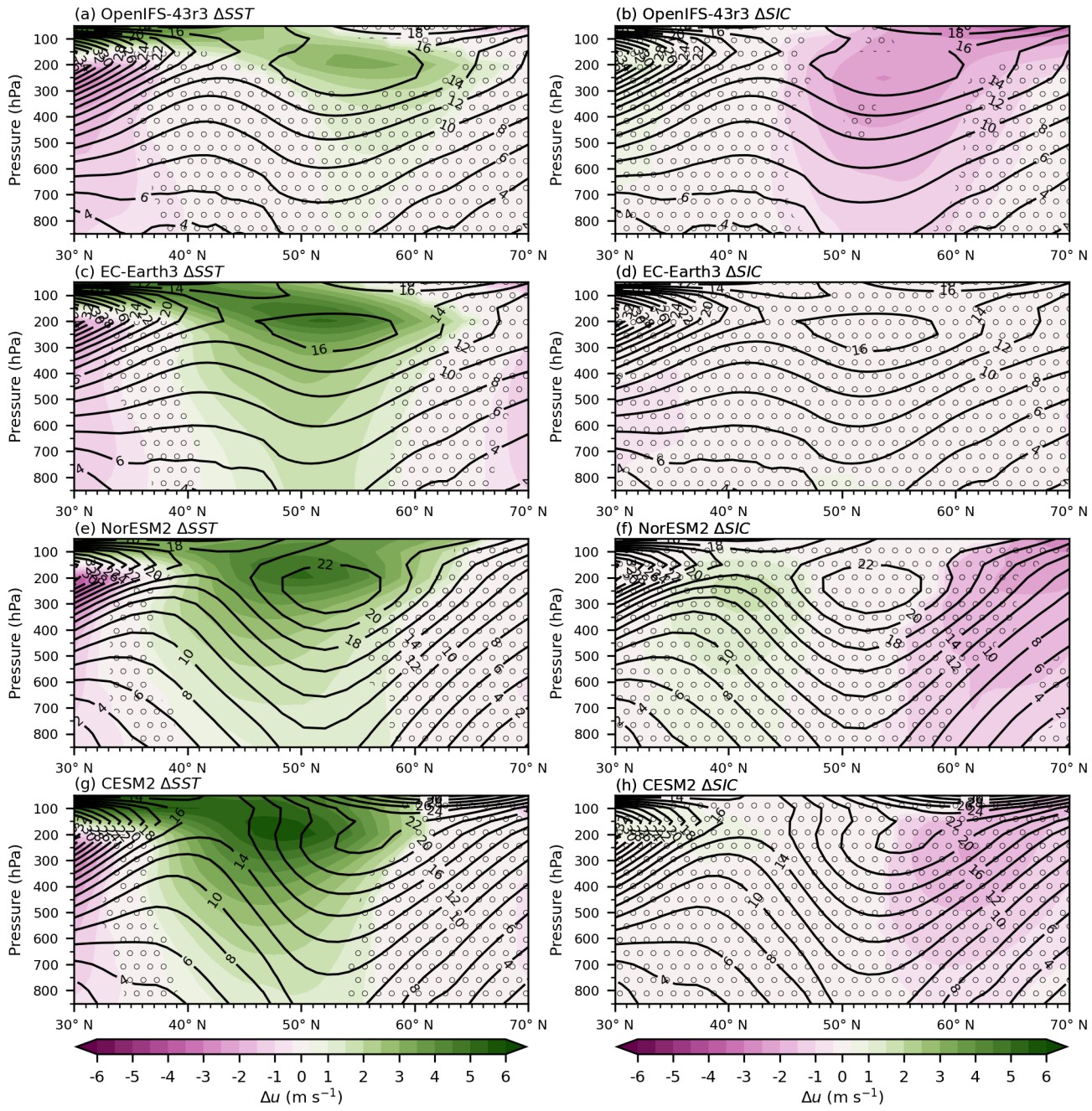

**Figure 8.** SST response $\Delta SST$ (a,c,e,g) and SIC response $\Delta SIC$ (b,d,f,h) of the zonal mean (15° W – 35° E) of the 39-year DJF mean of zonal wind speed $u$ (in colour shading), the Baseline climatology (in black contours) for (a) & (b) OpenIFS-43r3, (c) & (d) EC-Earth3, (e) & (f) NorEMS2, and (g) & (h) CESM2. Shading indicates as consistent sign according to CDR testing and stippling indicates statistically insignificant changes (FDR corrected t-test).

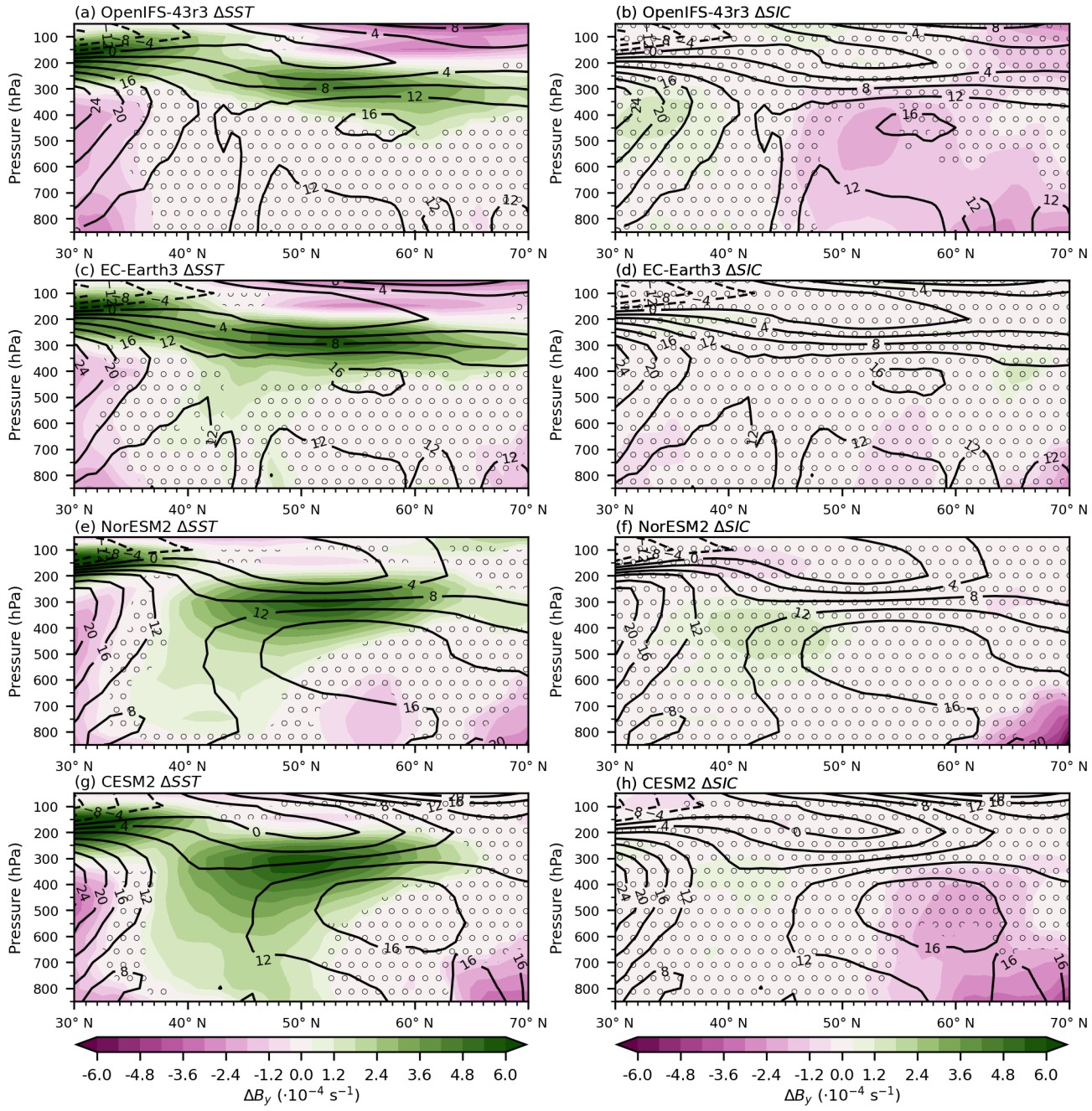

**Figure 9.** SST response $\Delta SST$ (a,c,e,g) and SIC response $\Delta SIC$ (b,d,f,h) of the zonal mean (15° W – 35° E) of the 39-year DJF mean of baroclinicity $B_y$ (in colour shading), the Baseline climatology (in black contours) for (a) & (b) OpenIFS-43r3, (c) & (d) EC-Earth3, (e) & (f) NorEMS2, and (g) & (h) CESM2. Shading indicates as consistent sign according to CDR testing and stippling indicates statistically insignificant changes (FDR corrected t-test).

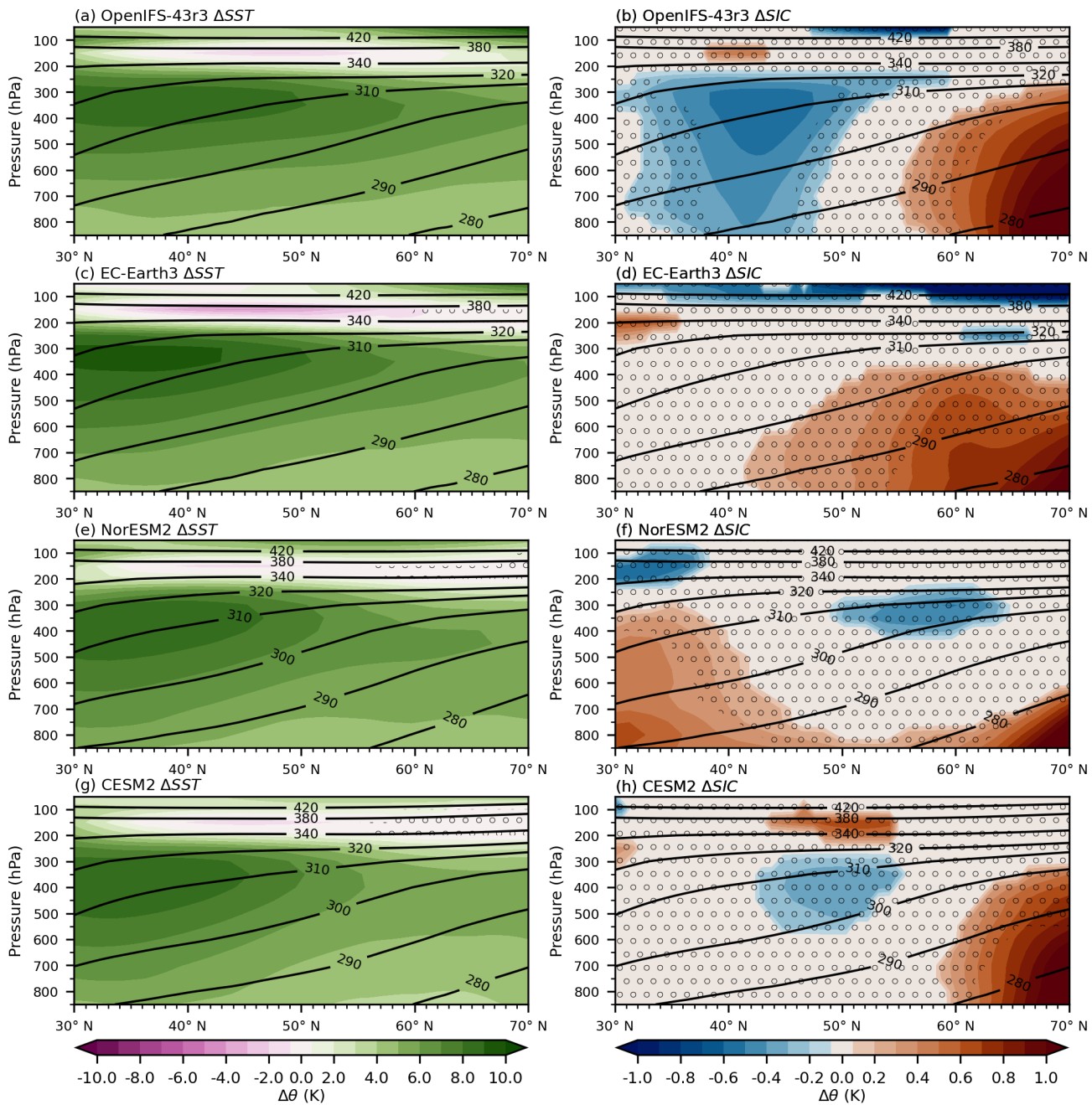

**Figure 10.** SST response $\Delta SST$ (a,c,e,g) and SIC response $\Delta SIC$ (b,d,f,h) of the zonal mean (15° W – 35° E) of the 39-year DJF mean of potential temperature $\theta$ (in colour shading), the Baseline climatology (in black contours) for (a) & (b) OpenIFS-43r3, (c) & (d) EC-Earth3, (e) & (f) NorEMS2, and (g) & (h) CESM2. Shading indicates as consistent sign according to CDR testing and stippling indicates statistically insignificant changes (FDR corrected t-test). Note that different colour scales are used because the absolute values of $\Delta SST$ and larger than those of $\Delta SIC$.

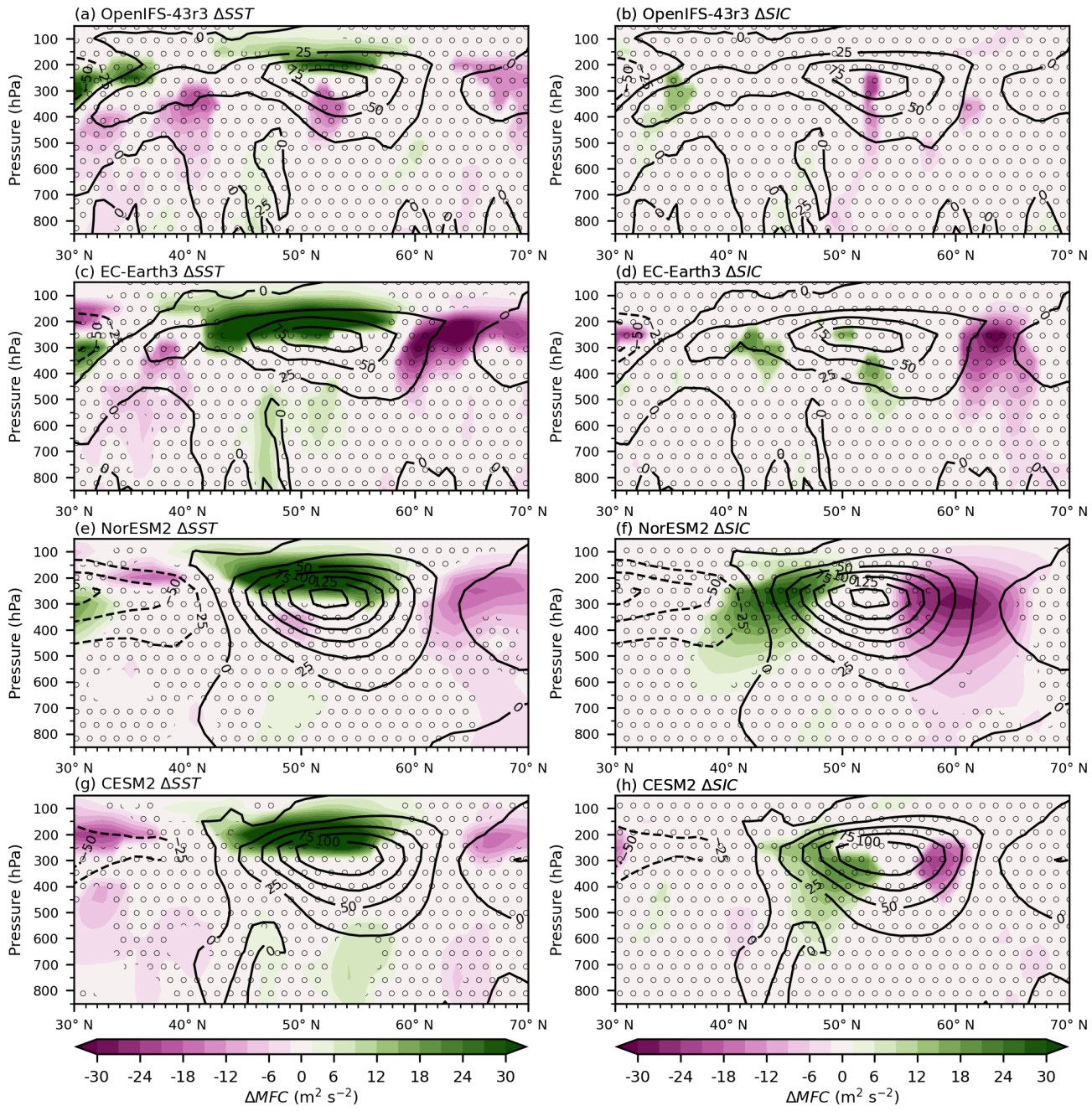

**Figure 11.** SST response $\Delta SST$ (a,c,e,g) and SIC response $\Delta SIC$ (b,d,f,h) of the zonal mean (15° W – 35° E) of the 39-year DJF mean of momentum flux convergence $MFC$ (in colour shading), the Baseline climatology (in black contours) for (a) & (b) OpenIFS-43r3, (c) & (d) EC-Earth3, (e) & (f) NorEMS2, and (g) & (h) CESM2. Shading indicates as consistent sign according to CDR testing and stippling indicates statistically insignificant changes (FDR corrected t-test).

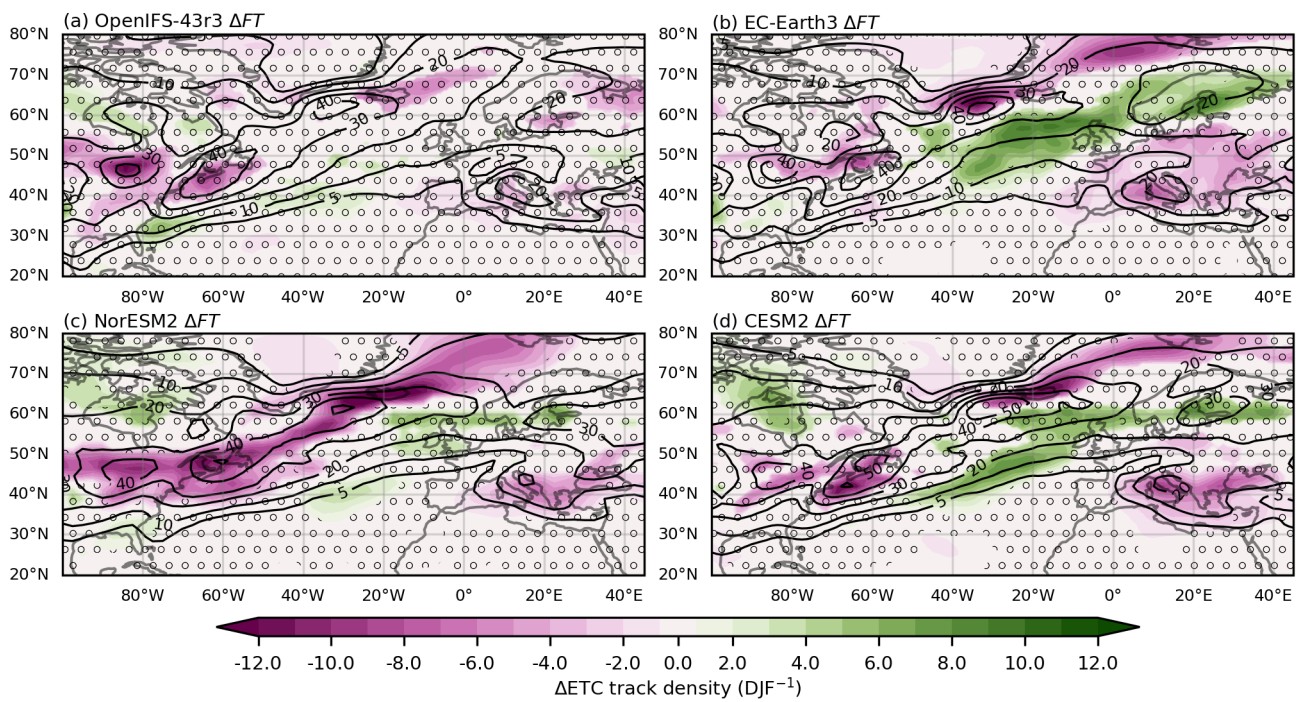

**Figure 12.** Future response $\Delta FT$ of the 39-year DJF mean ETC track density (in colour shading), the Baseline climatology (in black contours) for (a) OpenIFS-43r3, (b) EC-Earth3, (c) NorEMS2, and (d) CESM2. The unit is number of ETCs per $5°$ spherical cap per winter season (DJF). Shading indicates as consistent sign according to CDR testing and stippling indicates statistically insignificant changes (FDR corrected t-test).

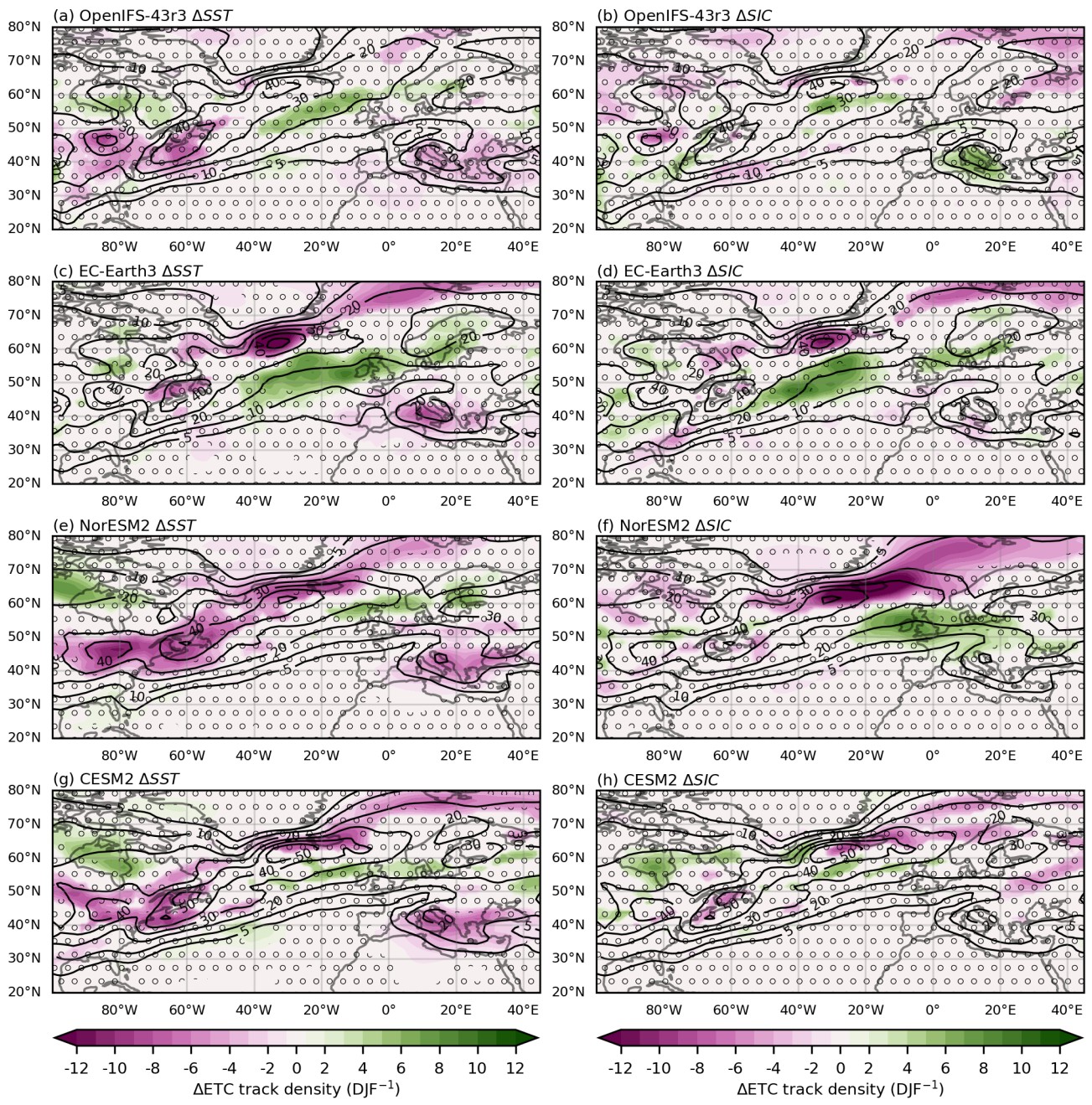

**Figure 13.** SST response $\Delta SST$ (a,c,e,g) and SIC response $\Delta SIC$ (b,d,f,h) of the 39-year DJF mean ETC track density (in colour shading), the Baseline climatology (in black contours) for (a) & (b) OpenIFS-43r3, (c) & (d) EC-Earth3, (e) & (f) NorEMS2, and (g) & (h) CESM2. The unit is the number of ETCs per $5°$ spherical cap per winter season (DJF). Shading indicates as consistent sign according to CDR testing and stippling indicates statistically insignificant changes (FDR corrected t-test).

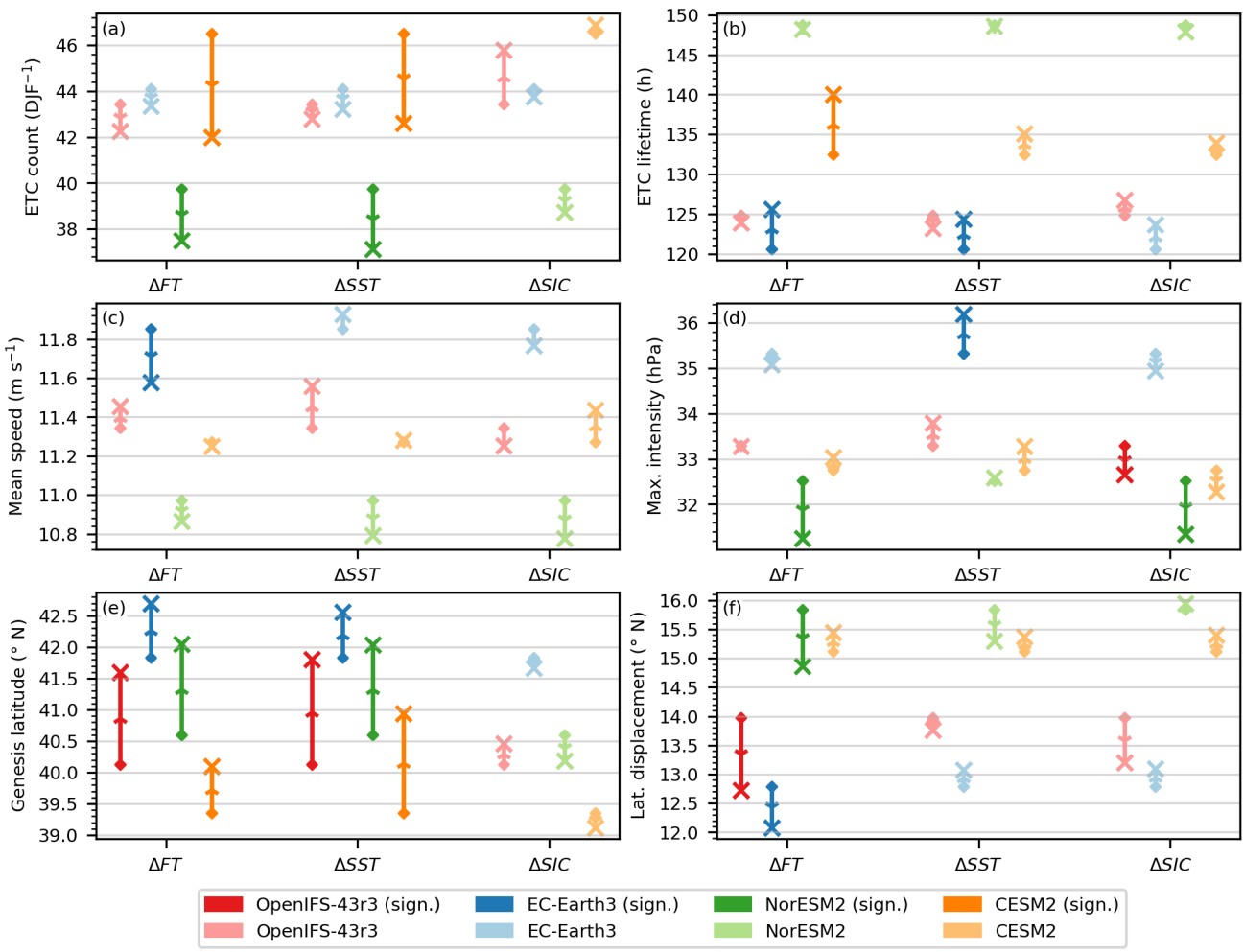

**Figure 14.** The change in mean values of ETC count per winter season DJF (a), ETC lifetime (b), mean speed (c), maximum intensity (d), genesis latitude (e), and latitudinal displacement (f) for OpenIFS-43r3 (red), EC-Earth3 (blue), NorESM2 (green), CESM2 (purple) with saturated colours indicating statistical significance (Mann-Whitney U test p-value < 0.05). Diamonds represent the Baseline simulation and crosses represent the corresponding experiment, respectively. Arrows are added to aid the interpretation of the direction from Baseline to the corresponding experiment.

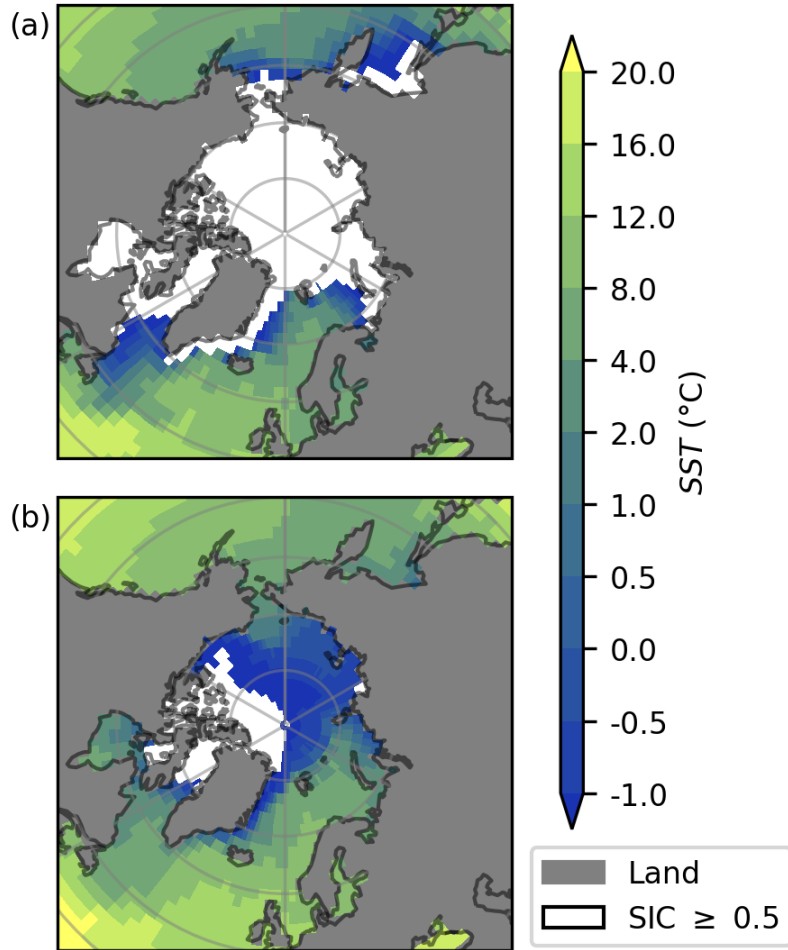

**Figure A1.** DJF mean of the boundary conditions of sea surface temperature $SST$ (shading) and sea ice cover $SIC$ (white where $SIC$ exceeds 0.5) for the Baseline simulations (a) and Future SSP 5-8.5 simulation (b).

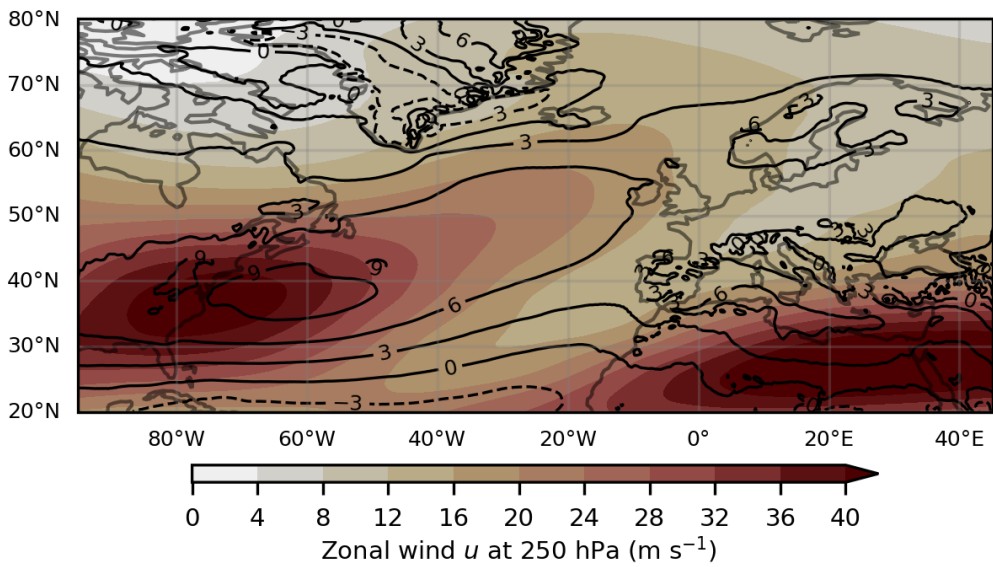

**Figure A2.** DJF mean of zonal wind speed $u$ in ERA5 reanalysis (years: 1950 − 1969). The 250 hPa level in colour shading and 850 hPa level in black contours (m s$^{-1}$).