# Peer review of "The future North Atlantic jet stream and storm track: relative contributions from sea ice and sea surface temperature changes"

_EGUsphere, 2024_

## Author Comment (AC1)

Responses to Reviewers' Comments for Manuscript

EGUSPHERE-2024-3713

**The future North Atlantic jet stream and storm track: relative contributions from sea ice and sea surface temperature changes**

Addressed Comments for Publication to

Weather and Climate Dynamics

by

Daniel Köhler, Petri Räisänen, Tuomas Naakka, Kalle Nordling, and

Victoria A. Sinclair

Dear Prof. Silvio Davolio,

Please find enclosed the revised version of our previous submission entitled "The future North Atlantic jet stream and storm track: relative contributions from sea ice and sea surface temperature changes" with manuscript number EGUSPHERE-2024-3713. We would like to thank you and the reviewers for the valuable comments which helped improving the quality of our manuscript. In this revision, we have carefully addressed the reviewers' comments. A summary of main modifications and a detailed point-by-point response to the comments from Reviewers 1 and 2 (following the reviewers' order in the decision letter) are given below.

Sincerely,

Daniel Köhler, Petri Räisänen, Tuomas Naakka, Kalle Nordling, and Victoria A. Sinclair

**Note:** To enhance the legibility of this response letter, all the editor's and reviewers' comments are typeset in boxes. Rephrased or added sentences are typeset in color. The respective parts in the manuscript are highlighted to indicate changes.

**Authors' Response to the Editor**

> **General Comments.** Your reference list includes works "in preparation". Such works can be cited upon submission if being available to the reviewers. They should not be cited in the final, accepted manuscript, unless published, accepted for publication, or available as preprint with a DOI.

**Response:** We appreciate your handling of the review process.

We updated the reference to include the available preprint with a DOI.

> **General Comments.** Please ensure that the colour schemes used in your maps and charts allow readers with colour vision deficiencies to correctly interpret your findings. Please check your figures using the Coblis – Color Blindness Simulator (https://www.color-blindness.com/coblis-color-blindness-simulator/) and revise the colour schemes accordingly with the next file upload request. -> Fig. 4

**Response:** We appreciate your handling of the review process and apologise for the inconsistency with the guidelines for accessibility.

> The colour schemes in Fig. 4 and Fig. 14 were adjusted to account for colour vision deficiency.

**Authors' Response to Reviewer 1**

**General Comments.** Using coordinated simulations with four different global atmospheric models, the study explores the roles of future sea surface temperature changes and sea-ice loss for changes of the wintertime North Atlantic jet streams and storm tracks. The simulated changes are extensively described with a focus on the dynamical drivers of the North Atlantic jet stream changes, namely the changes in the baroclinicity and momentum flux convergence. However, differences between the model responses with respect to the position and strength of jet stream and storm track shifts, point to uncertainties and leaving the European climate's future response uncertain. Given that there is still a debate within the scientific community about the impact of Arctic sea ice loss in the mid-latitude westerlies and storm tracks (with the possibility that Arctic sea ice loss weakens the North Atlantic jet stream favoring more severe cold winters), the topic of this study is timely and relevant. As reasons for the disagreement in the response to sea ice loss found in modelling studies include differences in the forcing fields (pattern and magnitude of forcing), coordinated experiments are an appropriate approach to tackle this question.The Polar Amplification Model Intercomparison Project (PAMIP, Smith et al., 2019) as part of CMIP6 provided a large set of coordinated experiments which allows to study the relative roles of local sea ice and remote sea surface temperature changes in response of the global climate system to changes in Polar sea ice.

This study is in my view complementary using a smaller set of models and different forcing fields. The manuscript is well-structured, but especially the part describing the results is lengthy, and, at some places, provides too much details. In comparison, the discussion and conclusion part is too short, and misses links to relevant other studies.Overall, the submitted manuscript needs careful and major revision.

**Response:** We thank the reviewer for their constructive comments and are pleased to hear that the reviewer finds our study timely and relevant. We have copied all the

reviewers comments below in blue shaded boxes and have added our detailed response to each comment in black text, any changes to the manuscript are set in gray boxes.

**Major comments**

> **Comment 1**
>
> I appreciate the careful description of the figures in the results sections. Nonetheless, I recommend a shortening of the description part of the manuscript, especially sections 4 to 6, to allow for clearer main messages. In addition, I strongly recommend more discussions of the results in comparison to other studies. This could be done either in the respective sections or as an additional section. As one example, I would like to mention the role of deep Arctic warming for mid-latitude atmospheric circulation changes which is discussed e.g. in Cohen et al. (2020), Xu et al. (2023), Kim et al. (2021). Since the model results presented here do show significant differences in the vertical extent of Arctic warming in response to sea-ice loss (figure 10), the results obtained here have to be discussed in the context of other studies which will certainly provide interesting insights! Such discussion are needed also for the other interesting results to place them in the context of other studies.

**Response:** We decided to streamline the presentation of the results. For example, in Section 4.1, where the Future response of the jet stream is presented, we have omitted the detailed description of the minor differences between EC-Earth3, NorESM2, and CESM2. This allows us to focus more on the scientifically relevant major differences between OpenIFS-43r3 and the other 3 models. Furthermore, several sentences which were repeating similar information were combined throughout Sections 4, 5, and 6. Next, explicitly mentioned numerical values were removed in places where they are easily and directly readable from the Figures. Previously introduced acronyms were used to shorten the text, enabling us to focus more on the results.

We added a Discussion section which discusses the results and places them in the context of existing literature:

- We address the robustness of the results against internal variability (Reviewer 1: Major comments: Comment 4).

- We also discuss the importance of the vertical extent of Arctic warming on the mid-latitude circulation responses (Labe et al., 2020; Xu et al., 2023).

- We discuss our results of the Future response of upper level jet stream compared with results from CMIP6 results (Harvey et al., 2020). We find good agreement between our results and CMIP6.

- We discuss that the results in the cross-section response are not directly comparable with previous studies, because they tend to focus on zonal means across the globe.

- We also discuss the changes to the storm tracks and ETCs, which are similar to previous studies Hay et al. (2023); Yu et al. (2023); Priestley and Catto (2022)

> **Comment 2**
>
> Based on the above mentioned extended discussions, the conclusions in section 7 have to be improved to highlight the new insights from this study and to elaborate on future research.

**Response:** Based on the reviewers comment, we improved the presentation of the novel insights by removing a few sentences, which we are discussing the results and are now incorporated in the new Discussion section, and combining sentences to better highlight the main results. We expanded on future research in the Conclusions section. Future research includes studying the impact of changed sea-ice and sea surface temperatures on extreme extratropical cyclones (ETC) and related extreme wind and precipitation events in the CRiceS simulations. Furthermore, it would be worth investigating more deeply the baroclinicity and eddy-driven mechanisms of the jet stream on an interannual timescale and connecting them to the activity of the storm tracks.

**Comment 3**

In the introduction, the benefits of coordinated model experiments should be discussed more in depth, in particular the PAMIP approach (Smith et al., 2019), and the differences to the approach applied in this study. It would be also good to explain a bit more the role of the coordinated model experiments for the whole EU project CriceS.

**Response:** We agree with the reviewer, and we elaborated more on the benefits of the coordinated model experiments conducted within CRiceS. For example, the stronger forcing of SST and SIC, which originate from one Earth system model rather than a multimodel mean. Moreover, the high output frequency of a wide range of atmospheric variables enable extensive analysis of physical mechanisms.

We also have provided more context of the simulations within the CRiceS project in the Introduction. CRiceS focusses on the role of the polar processes and their feedback loops in polar and global climate. These simulations contribute to quantifying processes that drive interactions and teleconnections between the higher and lower latitudes.

We have highlighted the differences between the CRiceS simulations and the PAMIP approach in Section 2.1. The key differences are the 40-year continuous simulation in CRiceS opposed to a 100-member ensemble of 1-year simulations in PAMIP. Another difference is the strength of the forcing, the CRiceS simulations impose SST and SIC changes according to the SSP 5-8.5 scenario in ACCESS-ESM1, which is equivalent to a global warming level of +4.4 K compared to Baseline simulation (1950–1969). For comparison, PAMIP forcing is equivalent to a +2 K global warming level. A Figure showing the sea surface temperature and sea-ice cover in the Baseline simulation and the Future simulation has been added to the Appendix.

Details on the benefits of the simulations and context were added to the Introduction of the manuscript. Key differences to PAMIP have been addressed in Section 2.1. A Figure was added showing the SST and SIC boundary conditions to the

Appendix.

**Comment 4**

The relative small sizes of the model runs with 40 years and what does this mean for the robustness of the results should be more critically discussed, given that, e.g. Peings et al. (2021) showed that even 100 years might be not enough to capture the remote atmospheric response to future Arctic sea ice loss against the large internal variability.

**Response:** We thank you the reviewer for this important comment and agree that a critical assessment is required. It should be noted in this response that although we have only 40-year simulations, the SST and sea-ice forcings are stronger than in the PAMIP experiments, which acts to boost the signal-to-noise ratio. We have improved our statistical testing (more details found in Reviewer 1: Major comments: Comment 5) by implementing the consistency discovery rate (CDR) measure for the response as proposed by Peings et al. (2021). The CDR testing together with the stronger forcing due to more sea-ice loss compared to PAMIP (Reviewer 1: Major comments: Comment 3) show that the presented responses have a consistent sign. Additionally, we have added some discussion of this issue in the newly added Discussion section.

**Comment 5**

The applied statistical testing has to be critically reviewed. For all testings, a two-sided t test has been applied. For some of the shown distributions (e.g. ETC lifetime in fig. 4b) it is obvious that the assumptions for the t-test, in particular normally distributed samples, are not fulfilled. For testing differences in fields, the tested fields are highly spatio-temporally correlated. Thus the t-test has to be controlled for field significance (e.g. by applying the false discovery rate (FDR; Wilks, 2016).

**Response:** We appreciate the reviewer bringing the weaknesses of the statistical testing to our attention. The primary interest is whether the Future, SST or SIC simulations tend to show higher or lower values in the ETC characteristics compared to the Baseline simulations, therefore we have replaced the t-test with the non-parametric Mann-Whitney U test for the ETC properties (Section 3.3, Figure 4) and the ETC response (Section 6.3, Figure 14). This did not affect the main conclusion about changes to the ETC properties. Furthermore, we applied the false discovery rate correction (FDR, described by Wilks (2016)) to fields with high spatio-temporal correlation (Figure 5, 6, 7, 8, 9, 10, 11, 12, 13). We added a short explanation of the false discovery correction in Section 2.5. The FDR correction did not impact the majority of the results. An important exception to this is that some responses to sea-ice loss in the zonal wind speed at 250 hPa, which we originally deemed significant. Especially the reduction of the zonal wind, $u$, over Europe in OpenIFS-43r3, turned out to be insignificant when the FDR correction is applied. However, the reduction of $u$ in the vertical cross-section over Europe remains significant.

Additionally, we evaluated the consistency discovery rate (CDR), a metric proposed by Peings et al. (2021), which provides information about the significance of the sign of the response. CDR performs a random subsampling without repetitions of 20 of the available 39 samples for the seasonal difference and calculates the sign of the mean response. This is repeated 1000 times. When 90% of the subsamples agree on the sign, the response is considered to have a significant/consistent sign. In the Figures in the revised manuscript,

coloured shading is only shown if the CDR test is significant. After applying CDR testing, there was little to no changes to the result. The affected results are the SIC response of potential temperature (Fig. 10b,d,f,g) and momentum flux convergence (Fig. 11b,d,f,g), visible by a reduction of the shared area. However, the additional CDR testing did not change any conclusions.

> We revised the significance testing process, which led to changes in the description of statistical testing (Sect. 2.5), updates in the significance statements in the results (Sect. 3.3, 4.2, 5.2 and 6.3) and in updates in Figures and their caption (Fig. 5, 6, 7, 8, 9, 10, 11, 12, 13).

**Comment 6**

I recommend an extended descriptions of the model experiments. I recognized that more information is given in Naakka et al. (2024), but it would help the reader to have more information also in this manuscript. For the description of the experiments, a table would be good. Since the model simulations are atmosphere-only simulations, I suggest to mention the similarities in the models, too: e.g. NorESM atmospheric component is based on CESM atmospheric component, and EC-Earth atmospheric component is based on an earlier version of OpenIFS. This should be also included into the discussion of the results.

**Response:** We thank the reviewer for the suggestion to expand on the descriptions of the experiments. We have added a Table summarising the experiments to Section 2.1. Moreover, we have added more details on the model set-up. For example, we provide the change in the global average near surface temperature in ACCESS-ESM1.5 (the model providing the SIC and SST forcing fields) between the CRiceS defined Baseline and Future periods.

In Section 2.1, we highlight the similarities of the atmospheric components amongst

the used models. As proposed, the connections between OpenIFS-43r3 and EC-Earth3 discussed, as well as the commonalities of NorESM2 and CESM2. Furthermore, we included into the Discussion section that OpenIFS-43r3 is a scientifically important outlier as it disagrees with the results from EC-Earth3 which is based on an earlier version of OpenIFS-43r3.

**Minor comments**

**Comment 1**

Introduction, L36-37: References are missing.

**Response:**

We have added the studies by Smith et al. (2022) and Screen et al. (2022) which address the connection between the zonal wind/jet stream response and the eddy feedback. Furthermore, we added citation for coordinated multi-model efforts like CMIP6 (Eyring et al., 2016), ScenarioMIP (O'Neill et al., 2016), and PAMIP (Smith et al., 2019) in the same paragraph.

**Comment 2**

Section 2, L112-115: Beside the E-vector, the Plumb flux (Plumb, 1985)and the localized EP flux (Trenberth, 1986) allows local diagnosis of the three-dimensional propagation of wave activity. Could you, please comment, why you have chosen the E-vectors?

**Response:** The advantage of the divergence of E-vector is that its formulation in the zonal average is identical to the Eliassen-Palm flux divergence (Trenberth, 1986). Its

meridional component was used in Smith et al. (2022) to assess the impact of eddies feeding back to the atmospheric mean flow and study the future changes to the zonal wind, which is closely related to the jet stream. Therefore, we decided it is an adequate metric to assess the mechanism of the eddy-driven jet, and it provides a degree of comparability to Smith et al. (2022).

> We added additional motivation for our choice in Section 2.3, as well as referring to Smith et al. (2022) using Eliassen-Palm flux to study the response of zonal wind speed to sea ice changes.

**Comment 3**

Section 2, L123-124: For frictionless motion!

**Response:** We have corrected the inaccuracy.

> We added 'for frictionless motion' to the text.

**Comment 4**

Section 3.1, L164-165: I would appreciate to see a corresponding figure to show the differences in zonal wind and baroclinicity to ERA5, e.g. in the appendix.

**Response:** We have included a Figure in the Appendix showing the climatological mean (1950–1969) zonal wind speed in ERA5 at 250 hPa and 850 hPa. The principal findings are that OpenIFS-43r3 and EC-Earth3 structurally agree well with ERA5 at 250 hPa and 850 hPa. However, representation of the jet stream at 850 hPa in NorESM2 and CESM2 is too zonal, and the maximum is located too far to the east in the North Atlantic.

We do not provide a plot of the baroclinicity from ERA5 as it is impractical for us to compute. To produce an equivalent Figure, we would have to download a large volume of ERA5 data at high temporal resolution over the same 20-year time period our baseline SSTs and sea ice were taken from. This is very time-consuming and requires a large amount of disk space, which is not easily available.

> A Figure showing the mean zonal wind speed in ERA5 at 250 hPa and 850 hPa for the time period 1950 – 1969 was added to the Appendix. We added a reference to it in Section 3.1.

Section 3.2, L199-200: Please explain, why the focus is on the jet exit region?

**Response:** We are focussing on Europe, as the impacts of changes to the jet and storm track will be experienced by millions of people. In our simulations, the jet exit region of the North Atlantic jet stream is located west of and over Europe. In particular, the left jet exit region is an area providing upper-level forcing inducing upward motion, which is favourable for the intensification of extratropical cyclones. Additionally, this particular region shows differences between models in the Baseline simulation and is of special interest due to the differing Future and SIC responses between models.

> We added the motivation for focussing on the jet exit region over Europe to beginning of Section 3.1.

Section 3.2, L220: Isn't the North-Atlantic jet an eddy-driven jet in all models?

**Response:** We would like to apologise for the inaccuracy. We corrected the statement to reflect our original intention that the contribution of the eddy driven mechanism is the largest in NorESM2 in the jet exit region over Europe.

> We replaced 'This emphasises the eddy-driven nature of the jet strength in NorESM2' with 'Of all four models, the jet stream in NorESM2 has the highest contribution from eddy momentum flux convergence.'

**Comment 7**

Section 3.3, L223: 'ETCs affecting Europe (30° N - 70° N, 15° W - 35° E),' Does that mean those ETC's which have their endpoint in that area?

**Response:** We appreciated this comment and have now provided clarification on L223 and on L437. Specifically, for ETCs to be included in this analysis, they have to remain within the area (30° N - 70° N, 15° W - 35° E) for at least 48 hours. This is regardless of where there genesis or lysis is.

> We added 'at least 48 hours within' to the brackets –> (at least 48 hours within 30° N – 70° N, 15° W – 35° E) on L223

**Comment 8**

Section 5.1, L353: 'to increase the jet speed located above the maximum in the Baseline simulation' For CESM, the increase in the jet speed is clearly southward of the maximum in the Baseline simulation, which is not so obvious in the other models (Fig. 8g). Could you, please comment on this?

**Response:** We apologise for the clearly wrong statement of the location of the jet speed increase in CESM2, as the maximum increase in jet speed for the Future and SST responses is located south of the Baseline maximum. To reflect the results shown in Figures 8g, and 9g, we have corrected the description of the location of the jet speed changes and connected it to the maximum increase in baroclinicity which is also located south of the Baseline maximum in CESM2.

> We corrected the description of the jet stream increase and mechanism for CESM2 in Section 5.1. The revised description reads as follows: "Unlike the other models, the change in baroclinicity is largest south of maximum in the Baseline simulation. Together with the change in $MFC$, this increases the jet speed located southward of the maximum in the Baseline simulation."

**Authors' Response to Reviewer 2**

**General Comments.** Summary: This paper examines the North Atlantic and European response of the jets and storm tracks across four atmospheric GCMS with sea ice, SSTs, or both together, as the surface forcing. The differences across the four models are discussed for their present day climatologies and for end their 21st century climate. The relative contributions of each SST and SIC make to the full response is examined. One model is an outlier for many of the responses. Overview: the paper is mostly well-written and clear, but is also too long and wordy in places. The figures are of good quality. I find the mechanistic approach to understanding inter-model differences to be illuminating and a real strength of this work compared to past ones. However, I have some concerns that the differences being discussed are an artefact of the short time period being used (39 winters), and the potential impact of seasonal-to-decadal atmospheric variability hampering a meaningful comparison between the models, as well as in their responses. As the experiments themselves are not the topic of the paper, I'll restrict my comments to suggestions for ways that the analysis presented and discussion around it could be framed to create a stronger paper. For example, there is too much detail in the results sections, and the end result is that the reader is left without a clear view on what are the important results. I'd suggest reframing the results around a few salient points rather than listing off all the results. Another suggestion is to show inter-model differences in the figures themselves, alongside the significance of the difference. This way only relevant differences are discussed, even if they are influenced by variability due to the short time series.

**Response:** We thank the reviewer for their constructive comments and are pleased to hear that the reviewer considers the mechanistic approach as a strength of our study. We have copied all the reviewers comments below in blue shaded boxes and have added our detailed response to each comment in black text, any changes to the manuscript are set in gray boxes.

**Major comments**

**Comment 1**

There are some references missing from the paragraph around line 25-30. E.g. Ronalds et al 2019, Hay et al 2022

**Response:** The missing references have been added.

We added Hay et al. (2022), Ronalds and Barnes (2019), Zappa et al. (2018), and Yu et al. (2024) to the introduction to present the uncertainties of future projection of the (North Atlantic) jet stream and highlight its connections to the storm track response.

**Comment 2**

L34: McKenna paper uses a more primitive model (intermediate complexity) which you might want to make clear, as it's not directly comparable with GCM results.

**Response:** We agree with the reviewer that it is important to highlight the use of an intermediate complexity model.

We provided the information of the usage of the intermediate complexity model McKenna et al. (2018) to the reader. We also added a sentence referencing the mid-latitude teleconnection to sea-ice loss (Screen, 2017) using UK Met Office Unified Model version 6.6.3, which is the atmospheric component of the model HadGEM2-ES.

**Comment 3**

Paragraph beginning L45 and Section 2.1: As this is not a new approach, there are missing references here, from early work on sea-ice loss AGCM experiments (e.g. Deser et al 2010), to similar but coupled experiments from McCusker et al 2017 and Oudar et al 2017. Finally, how these experiments differ from PAMIP and what is added by performing these needs more discussion. Here is where the discussion on how the short time series relative to noise and variability could be important. What is the change in surface mean temperature, SSTs, sea-ice loss and pattern of loss, and could these be relevant for understanding differences with PAMIP?

**Response:** We thank the reviewer for the comment. We have added many missing references in the Introduction and Section 2.1.

Additionally, we expanded on the benefits and key differences to PAMIP in the Introduction and Section 2.1. The key difference is the strength of the forcing, the CRiceS simulations impose SST and SIC changes according to the SSP 5-8.5 scenario in ACCESS-ESM1 (2080-2099), which equivalent to a global warming level of +4.4 K compared to the Baseline simulation (1950–1969). For comparison, PAMIP forcing is equivalent to a +2 K global warming level. We added a Figure depicting the sea surface temperature and sea-ice cover patterns for the Baseline and Future SSP 5-8.5 simulations to the Appendix. Furthermore, the dataset for each experiment consists of a 40-year continuous simulation in CRiceS opposed to 100-member ensemble of 1 year simulations in PAMIP.

In the newly added Discussion section, we acknowledge that the results need to be interpreted carefully, due to weak sea-ice response compared to internal variability in the midlatitudes. However, we are able to detect consistent responses due to SSTs and SIC according to the consistency discovery rate (Reviewer 2: Major comments: Comment 4), which is likely due to a stronger signal due to a stronger forcing.

**Comment 4**

L85: If you used ssp126, could you scale to combine and get 78 winters to boost sample size?

**Response:** The point of the reviewer is taken, however, is unclear on how the scaling should be done. Scaling the ssp126 and combining the Future simulations to boost the sample size requires multiple assumptions. An example would the linearity of the responses with respect to a local or global metric like sea ice loss or near surface temperature, which would serve as the scaling factor. Furthermore, in our understanding, scaling upwards the SSP126 response would also inflate the noise due to internal variability in the difference SSP126-baseline. These are additional uncertainties which would weaken the strength of this work, which is the mechanistic interpretation of the Future, SST and SIC responses across models.

To reduce the uncertainty related to the short time series, we have improved our statistical testing by implementing the consistency discovery rate metric as proposed by Peings et al. (2021). The method detects a consistent sign of responses by repeatedly subsampling randomly 20-year climatologies (1000 subsamples). If 90 % of the subsamples agree on the sign, the sign of the response is considered significant. Applying this method didn't, however, change our conclusions significantly.

**Comment 5**

Combine Sect 2.2 with 2.3 as it isn't really used as a standalone metric.

**Response:** We agree with the reviewer to combine these two sections.

Sections 2.2 and 2.3 were combined.

**Comment 6**

L126: What is meant by this last sentence?

**Response:** The signs of the change in baroclinicity and momentum flux convergence provide insight into their contributions to acceleration/deceleration of the zonal wind speed. The intended meaning of the sentence is that opposite signs in baroclinicity and momentum flux convergence would mean that their effect on the zonal wind speed compensate each other, while the same sign indicates that effect are both act to accelerate/decelerate the zonal wind.

> The sentence was reworded to clarify the intended meaning. 'This allows for a qualitative interpretation of compounding or opposing effect on the responses of the zonal wind due to sea ice and SST changes.' -> 'By comparing the sign of changes in baroclinicity and momentum flux convergence, a qualitative interpretation of compounding or opposing mechanistic effects on the responses of the zonal wind due to sea ice and SST changes is possible.'

**Comment 7**

Sect 2.5 could be combined with 2.1, and I'd leave the NL term out and just discuss it where relevant. There is also an inconsistent use of the nomenclature used here and terms like 'Future' and I'd suggest ensuring consistent use of one or the other throughout.

**Response:** We agree to streamline the manuscript by combining Sections 2.1 and 2.5 and removed the equation defining the non-linear term $\Delta NL$. When introducing the response metrics, we have retained that the sea-ice response and sea surface temperature response don't equal the Future response, and there is a non-linear effect of simultaneously changing SST and SIC. Additionally, we discuss non-linear interactions where necessary,

which is in particular for the responses of extratropical cyclones (ETCs) properties to changes of SIC and SST.

We ensured the consistency of the nomenclature. 'Future response' or $\Delta FT$ refers to the response to changed SIC and SST. 'SIC response' or $\Delta SIC$ refers to the response to changed SIC. 'SST response' or $\Delta SST$ refers to the response to changed SST.

**Comment 8**

Sect 3.3: are the ETC metrics calculated across the entire track of cyclones passing through the box, or just for the portion of the track within the box? It could change interpretation of the results.

**Response:** The ETC metrics are calculated across the entire track for each track that satisfies the condition of 'at least 48 hours within 30° N – 70° N, 15° W – 35° E'.

> We specified in the text (Sect. 3.3, Sect 6.3) that the properties are calculated across the entire track of the ETC if the ETC remained for at least 48 hours within the European box.

**Comment 9**

Sect 4.2 When discussing the contribution of SIC or SST to FT, taking a pattern correlation and then giving a % spatial variance explained by each can give a quantitative way of expressing the relative importance of each.

**Response:** We appreciate the comment of the reviewer. We agree that the % of spacial variance explained is a useful tool for expressing relative importance. However, it fails at capturing the compensating effects for the SIC and SST response in OpenIFS-43r3,

which are of particular interest. To properly address the relative importances, more advanced statistical techniques such as PCA, sparse PCA, or multiple linear regression are required, which are beyond the scope of this paper.
* * *
**Comment 10**

Sect 6.2 The non-linearity of the ETC response is quite interesting and a novel result. I'd suggest spending a bit of time discussing it.

**Response:** We thank the reviewer for highlighting a novel result of our study. We discuss the novel results, including the non-linearity of the ETC response in the newly added Discussion section. In particular, we set the Future response of ETCs in the context of CMIP6 (Priestley and Catto, 2022), and the responses due to SST and SIC changes are compared with results from PAMIP (Hay et al., 2023; Yu et al., 2023). Overall, our results structurally agree with previous findings. The novelty of our results is to quantify the contributions of SST and SIC changes to the Future response across models. However, there is large residual due to internal variability and non-linear interactions of SST and SIC, which are difficult to untangle. We suggest further research using methods like PCA and other more advanced statistical methods to study the interaction between SIC and SST on ETCs and the storm track.
* * *
**Comment 11**

Throughout: Discussion and placing results within context of existing literature is lacking.

**Response:** We agree with the reviewer's comment. We added a Discussion section which discusses the results and places them in the context of existing literature:

- We address the robustness of the results against internal variability (Reviewer 2:

Major comments: Comment 3 & 4).

- We discuss our results of the Future response of upper level jet stream compared with results from CMIP6 results (Harvey et al., 2020). We find good agreement between our results and CMIP6.

- We discuss that the results in the cross-section response are not directly comparable with previous studies, because they tend to focus on zonal means across the globe.

- We also discuss the changes to the storm tracks and ETCs (Reviewer 2: Major comments: Comment 10).

- We discussed is the importance of the vertical extent of Arctic warming to mid-latitude circulation responses (Labe et al., 2020; Xu et al., 2023) and the zonal wind speed response in idealised simulations (Kim et al., 2021).

**Minor comments**

**Comment 1**

L1 with -> from

**Response:**

We changed 'with' to 'from'.

**Comment 2**

Last two sentences of abstract are kind of saying the same thing

**Response:** We clarified the latter sentence to address all results (jet stream, storm track) to distinguish it from the previous sentence, which specifically covers the changes to extratropical cyclone properties. The added part is highlighted in bold font.

'The responses of extratropical cyclones to future sea-ice cover and sea surface temperatures do not exceed the inter-model climatological differences. Notable differences in the future response **of the jet stream and storm track** occur, and thus considerable uncertainty remains in how the European climate will respond to a warmer climate.'

**Comment 3**

L17 & elsewhere: majorly is an informal/slang word and shouldn't be used in a scientific paper

**Response:**

We replaced 'majorly' with 'strongly'.

**Comment 4**

L17: enhancing -> enhanced

**Response:** Thank you for the comment.

We changed 'enhancing' to 'enhanced'.

**Comment 5**

L22: Move 'together' to after 'jet stream

**Response:**

> We moved 'together' to after 'jet stream'.
* * *
**Comment 6**

L31 remove 'and' before necessitating

**Response:**

> We removed the 'and'.
* * *
**Comment 7**

Starting at L36 there seems to be a new paragraph as it is unrelated to part before.

**Response:**

> We have started a new paragraph to separate the thematics of the Arctic Oscillation response and the zonal wind speed response to sea-ice loss.
* * *
**Comment 8**

L47: with -> from

**Response:**

> We changed 'with' to 'from'.

**Comment 9**

L70 repeating 'atmosphere'

**Response:**

We changed the sentence to be more specific about the nature of the models (Earth system models: EC-Earth3, NorESM2, CESM2 and General circulation model OpenIFS-43r3) to avoid repeating 'atmospheric'.

**Comment 10**

L101-103: This sentence is hard to understand

**Response:** We simplified the structure of the sentence to improve the readability.

We changed 'Relevant for the jet stream is the meridional temperature gradient which is negative (e.g. temperature decreases towards the poles) in the troposphere, leading to the zonal wind increasing with height.' to 'Relevant for the jet stream is the negative meridional temperature gradient (i.e. temperature decreases towards the poles) in the troposphere, which leads to an increasing zonal wind with height.'

**Comment 11**

L124: 'are interpreted by the zonal mean denoted' -> 'in the zonal mean, denoted'

**Response:**

We changed 'are interpreted by the zonal mean denoted' to 'in the zonal mean, denoted'.

**Comment 12**

L140: was computed -> is computed

**Response:**

We changed 'was computed' to 'is computed'.

**Comment 13**

L190:is EC-Earth3 here a typo?

**Response:** We thank the reviewer for spotting this crucial typo.

We changed 'EC-Earth3' to 'CESM2'.

**Comment 14**

L211: You used both variable names and acronyms here and elsewhere, which makes the paper longer.

**Response:** We removed unnecessary doubling up of names and acronyms.

**Comment 15**

L227: remove last sentence

**Response:**

We deleted the sentence.

**Comment 16**

L243: 'significantly the most equatorward' is awkward wording

**Response:** We agree with the reviewer that the wording is awkward, and the wording was changed.

We changed 'Furthermore, CESM2 has significantly the most equatorward mean latitude of genesis of ETCs affecting Europe (39.4° N) among all four models. On the other extreme, EC-Earth3 has significantly the most poleward genesis (41.8° N). No significant difference is found between OpenIFS-43r3 (40.1° N) and NorESM2 (40.6° N).' –> 'Moreover, out of the four models, CESM2 has the most equatorward mean latitude of genesis of ETC affecting Europe (39.4 N) and EC-Earth3 the most poleward genesis location (41.8 N). These models differ significantly from OpenIFS-43r3 (40.1 N) and NorESM2 (40.6 N), the difference between the latter two models being insignificant.'.

**Comment 17**

L270: as EC-Earth -> to EC-Earth

**Response:**

We changed 'as EC-Earth' to 'to EC-Earth'.

**Comment 18**

L287: Repeating OpenIFS (Perhaps you might want to shorten the four variables name you use, for readability)

**Response:** Presumably the reviewer's suggestion to shorten the variable names, refers to the model names. We are of the opinion that reducing the model names further would reduce understanding, and it would be inconsistent with Naakka et al. (2024).

The repeating instances of OpenIFS-43r3 have been removed.

**Comment 19**

Section 5: I'd remove the whole paragraph to shorten the paper.

**Response:**

We removed the paragraph at the beginning of Section 5.

**References**

Eyring, V., S. Bony, G. A. Meehl, C. A. Senior, B. Stevens, R. J. Stouffer, and K. E. Taylor (May 2016). "Overview of the Coupled Model Intercomparison Project Phase 6 (CMIP6) experimental design and organization." English. In: *Geoscientific Model Development* 9(5), pp. 1937–1958. DOI: 10.5194/gmd-9-1937-2016.

Harvey, B. J., P. Cook, L. C. Shaffrey, and R. Schiemann (2020). "The Response of the Northern Hemisphere Storm Tracks and Jet Streams to Climate Change in the CMIP3, CMIP5, and CMIP6 Climate Models." en. In: *J. Geophys. Res. Atmos.* 125(23), e2020JD032701. DOI: 10.1029/2020JD032701.

Hay, S., P. J. Kushner, R. Blackport, K. E. McCusker, T. Oudar, L. Sun, M. England, C. Deser, J. A. Screen, and L. M. Polvani (2022). "Separating the Influences of Low-Latitude Warming and Sea Ice Loss on Northern Hemisphere Climate Change." EN. In: *Journal of Climate* 35(8), pp. 2327–2349. DOI: 10.1175/JCLI-D-21-0180.1.

Hay, S., M. D. K. Priestley, H. Yu, J. L. Catto, and J. A. Screen (2023). "The Effect of Arctic Sea-Ice Loss on Extratropical Cyclones." en. In: *Geophys. Res. Lett.* 50(17), e2023GL102840. DOI: 10.1029/2023GL102840.

Kim, D., S. M. Kang, T. M. Merlis, and Y. Shin (2021). "Atmospheric Circulation Sensitivity to Changes in the Vertical Structure of Polar Warming." In: *Geophysical Research Letters* 48(19), e2021GL094726. DOI: 10.1029/2021GL094726.

Labe, Z., Y. Peings, and G. Magnusdottir (2020). "Warm Arctic, Cold Siberia Pattern: Role of Full Arctic Amplification Versus Sea Ice Loss Alone." en. In: *Geophys. Res. Lett.* 47(17), e2020GL088583. DOI: 10.1029/2020GL088583.

McKenna, C. M., T. J. Bracegirdle, E. F. Shuckburgh, P. H. Haynes, and M. M. Joshi (2018). "Arctic Sea Ice Loss in Different Regions Leads to Contrasting Northern Hemisphere Impacts." en. In: *Geophys. Res. Lett.* 45(2), pp. 945–954. DOI: 10.1002/2017GL076433.

Naakka, T., D. Köhler, K. Nordling, P. Räisänen, M. Tronstad Lund, R. Makkonen, J. Merikanto, B. H. Samset, V. A. Sinclair, J. L. Thomas, and A. M. L. Ekman

(2024). "Polar winter climate change: strong local effects from sea ice loss, widespread consequences from warming seas." [preprint]. DOI: 10.5194/egusphere-2024-3458.

O'Neill, B. C., C. Tebaldi, D. P. van Vuuren, V. Eyring, P. Friedlingstein, G. Hurtt, R. Knutti, E. Kriegler, J.-F. Lamarque, J. Lowe, G. A. Meehl, R. Moss, K. Riahi, and B. M. Sanderson (2016). "The Scenario Model Intercomparison Project (ScenarioMIP) for CMIP6." English. In: *Geoscientific Model Development* 9(9), pp. 3461–3482. DOI: 10.5194/gmd-9-3461-2016.

Peings, Y., Z. M. Labe, and G. Magnusdottir (2021). "Are 100 Ensemble Members Enough to Capture the Remote Atmospheric Response to +2°C Arctic Sea Ice Loss?" EN. In: *Journal of Climate* 34(10), pp. 3751–3769. DOI: 10.1175/JCLI-D-20-0613.1.

Priestley, M. D. K. and J. L. Catto (2022). "Future changes in the extratropical storm tracks and cyclone intensity, wind speed, and structure." English. In: *Weather and Climate Dynamics* 3(1), pp. 337–360. DOI: 10.5194/wcd-3-337-2022.

Ronalds, B. and E. A. Barnes (2019). "A Role for Barotropic Eddy–Mean Flow Feedbacks in the Zonal Wind Response to Sea Ice Loss and Arctic Amplification." In: *Journal of Climate* 32(21), pp. 7469–7481. DOI: 10.1175/JCLI-D-19-0157.1.

Screen, J. A. (2017). "The missing Northern European winter cooling response to Arctic sea ice loss." en. In: *Nat Commun* 8(1), p. 14603. DOI: 10.1038/ncomms14603.

Screen, J. A., R. Eade, D. M. Smith, S. Thomson, and H. Yu (2022). "Net Equatorward Shift of the Jet Streams When the Contribution From Sea-Ice Loss Is Constrained by Observed Eddy Feedback." en. In: *Geophys. Res. Lett.* 49(23), e2022GL100523. DOI: 10.1029/2022GL100523.

Smith, D. M., R. Eade, M. B. Andrews, H. Ayres, A. Clark, S. Chripko, C. Deser, N. J. Dunstone, J. García-Serrano, G. Gastineau, et al. (2022). "Robust but weak winter atmospheric circulation response to future Arctic sea ice loss." en. In: *Nature Communications* 13(1), p. 727. DOI: 10.1038/s41467-022-28283-y.

Smith, D. M., J. A. Screen, C. Deser, J. Cohen, J. C. Fyfe, J. García-Serrano, T. Jung, V. Kattsov, D. Matei, R. Msadek, Y. Peings, M. Sigmond, J. Ukita, J.-H. Yoon, and X. Zhang (2019). "The Polar Amplification Model Intercomparison Project

(PAMIP) contribution to CMIP6: investigating the causes and consequences of polar amplification." English. In: *Geoscientific Model Development* 12(3), pp. 1139–1164. DOI: 10.5194/gmd-12-1139-2019.

Trenberth, K. E. (1986). "An Assessment of the Impact of Transient Eddies on the Zonal Flow during a Blocking Episode Using Localized Eliassen-Palm Flux Diagnostics." EN. In: *Journal of the Atmospheric Sciences* 43(19), pp. 2070–2087. DOI: 10.1175/1520-0469(1986)043<2070:AAOTIO>2.0.CO;2.

Wilks, D. S. (2016). ""The Stippling Shows Statistically Significant Grid Points": How Research Results are Routinely Overstated and Overinterpreted, and What to Do about It." EN. In: *BAMS* 97(12), pp. 2263–2273. DOI: 10.1175/BAMS-D-15-00267.1.

Xu, M., W. Tian, J. Zhang, J. A. Screen, C. Zhang, and Z. Wang (2023). "Important role of stratosphere-troposphere coupling in the Arctic mid-to-upper tropospheric warming in response to sea-ice loss." In: *npj Climate and Atmospheric Science* 6(1), pp. 1–9. DOI: 10.1038/s41612-023-00333-2.

Yu, H., J. A. Screen, S. Hay, J. L. Catto, and M. Xu (2023). "Winter Precipitation Responses to Projected Arctic Sea Ice Loss and Global Ocean Warming and Their Opposing Influences over the Northeast Atlantic Region." EN. In: *J. Climate* 36(15), pp. 4951–4966. DOI: 10.1175/JCLI-D-22-0774.1.

Yu, H., J. A. Screen, M. Xu, S. Hay, and J. L. Catto (2024). "Comparing the Atmospheric Responses to Reduced Arctic Sea Ice, a Warmer Ocean, and Increased CO2 and Their Contributions to Projected Change at 2°C Global Warming." In: *Journal of Climate* 37(23), pp. 6367–6380. DOI: 10.1175/JCLI-D-24-0104.1.

Zappa, G., F. Pithan, and T. G. Shepherd (2018). "Multimodel Evidence for an Atmospheric Circulation Response to Arctic Sea Ice Loss in the CMIP5 Future Projections." en. In: *Geophysical Research Letters* 45(2), pp. 1011–1019. DOI: 10.1002/2017GL076096.

---

## Author Response (AR2)

Responses to Reviewers' Comments for Manuscript

EGUSPHERE-2024-3713

**The future North Atlantic jet stream and storm track: relative contributions from sea ice and sea surface temperature changes**

Addressed Comments for Publication to

Weather and Climate Dynamics

by

Daniel Köhler, Petri Räisänen, Tuomas Naakka, Kalle Nordling, and

Victoria A. Sinclair

Dear Prof. Silvio Davolio,

Please find enclosed the revised version of our previous submission entitled "The future North Atlantic jet stream and storm track: relative contributions from sea ice and sea surface temperature changes" with manuscript number EGUSPHERE-2024-3713. We would like to thank you and the reviewers for the valuable comments which helped improving the quality of our manuscript. In this revision, we have carefully addressed the reviewers' comments. A detailed point-by-point response to the comments from Reviewers 1 and 2 (following the reviewer numbers in the MS records) are given below.

We would like to bring to the editor's attention that we have modified the acknowledgments during this revision. We added, 'We thank the two anonymous reviewers for their constructive comments that helped to improve the manuscript.'

Sincerely,

Daniel Köhler, Petri Räisänen, Tuomas Naakka, Kalle Nordling, and Victoria A. Sinclair

**Note:** To enhance the legibility of this response letter, all the editor's and reviewers' comments are typeset in boxes.

**Authors' Response to the Editor**

> **General Comments.** Please ensure that the colour schemes used in your maps and charts allow readers with colour vision deficiencies to correctly interpret your findings. Please check your figures using the Coblis – Color Blindness Simulator (https://www.color-blindness.com/coblis-color-blindness-simulator/) and revise the colour schemes accordingly with the next file upload request. -> Fig. 4

**Response:** We appreciate your handling of the review process and apologise for the inconsistency with the guidelines for accessibility.

The colour schemes in Fig. 4 and Fig. 14 were adjusted to account for colour vision deficiency. Additionally markers were added in Fig. 4.

**Authors' Response to Reviewer 1**

> **General Comments.** Review of the manuscript "The future North Atlantic jet stream and storm track: relative contributions from sea ice and sea surface temperature changes" Daniel Köhler, Petri Räisänen, Tuomas Naakka, Kalle Nordling, and Victoria A. Sinclair submitted to WCD (WCD-2024-3713)
>
> I very much appreciate the efforts made by the authors to improve the manuscript, and the very careful and detailed responses to my comments from the last round of review. In particular i value the improved tests of the statistical significance and the related discussions, the shortening of the descriptive part of the manuscript, the inclusion of the discussion section 7 and the improved conclusions.

**Response:** We thank the reviewer for their constructive comments and are pleased to hear that the reviewer is positive about the improvements to the manuscript.

**Minor comments**

> **Comment 1**
>
> Although the the descriptive part of the manuscript (sections 4 to 6) has been shortened, it is still long. I would like to ask the authors to prove further shortening, in particular in section 6.

**Response:** We tried very hard to shorten section 6, however, it proved to be very difficult to do without omitting results, which in our opinion, are of interest to the scientific community. Originally, reviewer 2 also asked us to shorten the manuscript and is now satisfied with the length of results section after the first revision.

We have shortened section 6 by 4 lines, which results in the draft format of section 6 being now only just over 2 pages long.

> **Comment 2**
>
> L44-46: "Smith et al. (2022) showed that models tend to underestimate responses of mid-latitude tropospheric zonal wind due to changes in SIC when constrained by observations of the eddy momentum feedback." Please check this, i understood from the paper, that by using the observed eddy momentum feedback (which is larger than in the climate models) to scale the zonal wind responses in the models results in an increased ensemble mean zonal wind response.

**Response:** We thank the reviewer for pointing out the inconsistency with Smith et al. (2022).

We reformulated the sentence to reflect the intended meaning - that the tropospheric zonal wind response in climate model is likely to be underestimated as the eddy momentum feedback is too weak in climate models compared to observations.

We changed 'Smith et al. (2022) showed that models tend to underestimate responses of mid-latitude tropospheric zonal wind due to changes in SIC when constrained by observations of the eddy momentum feedback.' to 'Smith et al. (2022) showed that models tend to underestimate responses of mid-latitude tropospheric zonal wind due to changes in SIC as the eddy momentum feedback is too weak in climate models compared to observations.'.

> **Comment 3**
>
> L63: I suggest to also include Ogawa et al., GRL (2018) https://doi.org/10.1002/2017GL076502.

**Response:** We have added Ogawa et al. (2018) in the citation of previous studies conducting simulations with prescribed SST/SIC.

> **Comment 4**
>
> Table 1: I appreciate the table and respective description, but i suggest to skip the SSP 1-2.6 simulations in the table and the text, since they have not been analyzed in the manuscript.

**Response:** We appreciate the comment of the reviewer, and understand the suggestion to streamline the table and respective description. However, we consider the information about full extent of simulation data set important. It helps to motivate why the study focusses on an extreme warming scenario. The main reason is to allow for an improved detection of sea-ice loss signals. Additionally, the table and text inform the reader about the availability of a data set with a scenario similar to the PAMIP.

> **Comment 5**
>
> L684-685: sentence is not complete.

**Response:** Thank you for pointing this out. We changed 'Moreover, as this study investigates the mean changes to extratropical cyclones, further work on extreme cyclones and associated extreme events of wind and precipitation.' to 'Moreover, as this study investigates the mean changes to extratropical cyclones, further work on extreme cyclones and associated extreme events of wind and precipitation **is necessary**.'

**Authors' Response to Reviewer 2**

> **General Comments.** I am happy with the work the authors have put in to revising their manuscript. In particular, the efforts made to place this work in to better context and to strengthen the statistical testing are big improvements. I have very minor, mostly only technical comments, otherwise I am happy for this manuscript to be published.

**Response:** We thank the reviewer for their comments and are pleased to hear that the reviewer is positive about the improvements to the manuscript.

**Minor comments**

**Comment 1**

L69-70. Unclear. What do you mean by 'current' uncertainties? It is also not clear what 'This' at the start of the next sentence refers to. 'us' missing after permits.

**Response:** We deleted the word 'current'. The following sentence starting in 'This' was reformulated to reflect that studying the underlying physical mechanisms are enabled by the high frequency output of a wide selection of variables.

We changed 'This is enabled by high output frequency of a wide selection of atmospheric variables, which permits to examine the underlying physical mechanisms and identify structural differences in physical mechanisms of the response to projected SIC and SST across the selected models,' to 'Enabled by high output frequency of a wide selection of atmospheric variables, the CRiceS simulations permit to examine the underlying physical mechanisms and identify structural differences in physical mechanisms of the response to projected SIC and SST across the selected models.'.

**Comment 2**

L112. Have showed -> either have shown or showed

**Response:** We changed 'have showed' to 'have shown'.

**Comment 3**

L139-140. It's been shown before that not changing SSTs where ice is lost has a minor effect (Singarayer 2006, Deser et al 2010), and it also sounds like the resulting non-linear interactions are arising from that, but I don't think that's what the authors mean to imply.

**Response:** We clarified that the non-linear interaction arise from changing SSTs and sea ice cover simultaneously. We also inform the reader that the lacking increase in SSTs where sea ice is remove is a minor effect contributing to the differences between $\Delta FT$ and $\Delta SST + \Delta SIC$. We included references to earlier studies (Singarayer et al., 2006; Deser et al., 2010).

We changed 'The differences arise from the lack of the effect of changed SSTs where sea ice is removed and resulting non-linear interactions' to 'The differences arise from the non-linear interactions when SIC and SSTs are changed simultaneously. Furthermore, in the SIC simulations, when sea ice is removed SSTs are not changed. The lacking change in SSTs in these locations contributes to the differences, but it has been shown to be a minor effect (Singarayer et al., 2006; Deser et al., 2010).'.

**Comment 4**

L150. no need to capitalise Figures.

**Response:** We changed 'Figures' to 'figures'.

> **Comment 5**
>
> L330: I think there's a word missing after DeltaSST

**Response:** The sentence refers to the SST and SIC response in OpenIFS-43r3. The SIC response was missing. We changed 'The $\Delta SST$ of zonal wind speed cross-section from the OpenIFS-43r3 simulations are shown in Figure 8a.' to 'The $\Delta SST$ and $\Delta SIC$ of zonal wind speed cross-section from the OpenIFS-43r3 simulations are shown in Figure 8a and 8b.'.

> **Comment 6**
>
> L336. 'of jet core' -> of the jet core

**Response:** We changed 'of jet core' to 'of the jet core'.

> **Comment 7**
>
> L393: 'does not show any significant changes' -> is not significant

**Response:** We changed 'does not show any significant changes' to 'is not significant'.

> **Comment 8**
>
> L398. Last sentence is repetitive

**Response:** We deleted the repetitive sentence.

**Comment 9**

L399 & elsewhere: I think 'dominant' rather than 'dominating' is what is meant.

**Response:** We changed 'dominating' to 'dominant'.

**Comment 10**

L437. where -> with

**Response:** We changed 'where' to 'with'.

**Comment 11**

L473. delete 'is' between which and corresponds.

**Response:** We deleted the 'is' between which and corresponds.

**Comment 12**

L594 unfinished sentence

**Response:** Thank you for pointing this out. We changed 'Moreover, as this study investigates the mean changes to extratropical cyclones, further work on extreme cyclones and associated extreme events of wind and precipitation.' to 'Moreover, as this study investigates the mean changes to extratropical cyclones, further work on extreme cyclones and associated extreme events of wind and precipitation **is necessary**.'

**References**

Deser, C., R. Tomas, M. Alexander, and D. Lawrence (2010). "The Seasonal Atmospheric Response to Projected Arctic Sea Ice Loss in the Late Twenty-First Century." EN. In: *Journal of Climate* 23(2), pp. 333–351. DOI: 10.1175/2009JCLI3053.1.

Ogawa, F., N. Keenlyside, Y. Gao, T. Koenigk, S. Yang, L. Suo, T. Wang, G. Gastineau, T. Nakamura, H. N. Cheung, N.-E. Omrani, J. Ukita, and V. Semenov (2018). "Evaluating Impacts of Recent Arctic Sea Ice Loss on the Northern Hemisphere Winter Climate Change." en. In: *Geophys. Res. Lett.* 45(7), pp. 3255–3263. DOI: 10.1002/2017GL076502.

Singarayer, J. S., J. L. Bamber, and P. J. Valdes (2006). "Twenty-First-Century Climate Impacts from a Declining Arctic Sea Ice Cover." In: *J. Climate* 19(7), pp. 1109–1125. DOI: 10.1175/JCLI3649.1.

Smith, D. M., R. Eade, M. B. Andrews, H. Ayres, A. Clark, S. Chripko, C. Deser, N. J. Dunstone, J. García-Serrano, G. Gastineau, et al. (2022). "Robust but weak winter atmospheric circulation response to future Arctic sea ice loss." en. In: *Nature Communications* 13(1), p. 727. DOI: 10.1038/s41467-022-28283-y.